# Reconstructing ocean carbon storage with CMIP6 models and synthetic Argo observations

Katherine E. Turner[1], Doug M. Smith[2], Anna Katavouta[1,3], and Richard G. Williams[1]

[1]Department of Earth, Ocean, and Ecological Sciences, School of Environmental Sciences University of Liverpool, Liverpool, United Kingdom
[2]UK Met Office, Exeter, United Kingdom
[3]National Oceanography Centre, Liverpool, United Kingdom

**Correspondence:** Katherine E. Turner (K.E.Turner2@liverpool.ac.uk)

**Abstract.** The ocean carbon store plays a vital role in setting the carbon response to emissions and variability in the carbon cycle. However, due to the ocean's strong regional and temporal variability, sparse carbon observations limit our understanding of historical carbon changes. Ocean temperature and salinity profiles are more wide-spread and rapidly expanding due to autonomous programmes, and so we explore how temperature and salinity profiles can provide information to reconstruct ocean carbon inventories with Ensemble Optimal Interpolation. Here, Ensemble Optimal Interpolation is used to reconstruct ocean carbon using synthetic Argo temperature and salinity observations, with examples for both the top 100m and top 2000m carbon inventories. When considering reconstructions of the top 100m carbon inventory, coherent relationships between upper-ocean carbon, temperature, salinity, and atmospheric $CO_2$ result in optimal solutions that reflect the controls of undersaturation, solubility, and alkalinity. Out-of-sample reconstructions of the top 100m show that, in most regions, the trend in ocean carbon and over 60% of detrended variability can be reconstructed using local temperature and salinity measurements, with only small changes when considering synthetic profiles consistent with irregular Argo sampling. Extending the method to reconstruct the upper 2000m reveals that model uncertainties at depth limit the reconstruction skill. The impact of these uncertainties on reconstructing the carbon inventory over the upper 2000m is small, and full reconstructions with historical Argo locations show that the method can reconstruct regional interannual and decadal variability. Hence, optimal interpolation based on model relationships combined with hydrographic measurements can provide valuable information about global ocean carbon inventory changes.

## 1 Introduction

The global ocean plays a important role in the carbon cycle, being both a major reservoir of carbon and substantial sink of anthropogenic carbon transferred from geological storage to the coupled atmosphere-ocean-terrestrial system. The ocean is estimated to have taken up around 26% of anthropogenic emissions from fossil fuels and land use changes since 1850 (Khatiwala et al., 2013; DeVries, 2014; Terhaar et al., 2022; Friedlingstein et al., 2022). In addition to the long-term uptake of anthropogenic carbon, ocean carbon uptake exhibits interannual and decadal variability on global and regional scales (Landschützer et al., 2016; McKinley et al., 2017; Gruber et al., 2019b; McKinley et al., 2020). This variability impacts the ability to detect

trends in observations of the partial pressure of $CO_2$ in seawater (McKinley et al., 2016) and, through oceanic transport, can
lead to regionally enhanced acidification (Burger et al., 2020; Hauri et al., 2021). The ocean carbon inventory is thus an important integral measure of climate change, and up-to-date estimates on its behaviour are vital for understanding its evolution and impacts on the climate system.

In order to characterise the ocean carbon system with its strong regional and temporal variability, extensive spatial and temporal coverage in observations are required. Whereas there are millions of surface $pCO_2$ measurements to provide a surface view of the ocean carbon system and air-sea carbon fluxes (Bakker et al., 2016), complementary observations of ocean interior dissolved inorganic carbon (DIC) are limited by the logistics of ship-based bottle measurements. Campaigns for ocean interior DIC observations use repeat transects to provide high-quality observations for many regions (Sloyan et al., 2019). The transects resolve spatial variations in carbon but are limited in their ability to resolve temporal variability as they are repeated on decadal timescales. Alternatively, ocean DIC time series such as those found at the Bermuda Atlantic Time Series and the Hawaii Ocean Timeseries can resolve seasonal, interannual, and decadal variability, but are limited in how well they represent variability on larger spatial scales (Bates et al., 2014). The Bio-Argo programme has allowed for more autonomous sampling of ocean interior carbonate system variables such as pH (Claustre et al., 2020), from which interior DIC can be estimated; currently, however, autonomous measurement methods for interior DIC remain in development.

By itself, the sparsity of interior observations hinders the ability to produce a coherent global picture of recent ocean carbon changes from a storage perspective. A mapping technique is necessary to expand points of observations into more coherent spatial patterns of change and behaviour. Mapping techniques employ statistics to propagate information from observations to unobserved regions. Time-invariant climatologies of ocean carbon have been created using mapping procedure on data from repeat transects and other ship-based observations (Lauvset et al., 2021). Non-linear machine learning procedures have been able to reconstruct spatial patterns in interannual and decadal variability for surface $pCO_2$ (Landschützer et al., 2016; Landschützer et al., 2019; Gloege et al., 2022) as well as the seasonal cycle for DIC (Keppler et al., 2020), but the non-linearities can create substantial biases for regions with sparse observations (Bushinsky et al., 2019). Data assimilative methods that use observations to constrain model physics in forward experiments have also been expanded to include biogeochemistry (Verdy and Mazloff, 2017; Carroll et al., 2020). Multiple linear regression models have also been used to reconstruct changes in the anthropogenic carbon pool (Clement and Gruber, 2018; Gruber et al., 2019a).

For any mapping or data assimilative technique, it is imperative to use accurate statistics to avoid the erroneous propagation of information from observations. These statistical relationships can be parameterised or calculated from observations; however, parameterised statistics fail to reflect regional differences in the ocean, and for poorly observed variables, averaging over large length or timescales can lead to overly coarse covariance fields. State-of-the-art climate models provide complete pseudo-data in both space and time and therefore may be used to calculate these statistics, though model biases may lead to errors in the covariances. Nevertheless, fully-coupled climate models have been used to reconstruct ocean heat content trends and variability from observations, with clear improvements in recent years due to the expansion of the Argo programme (Smith and Murphy, 2007; Smith et al., 2015; Cheng and Zhu, 2016).

Ocean carbonate chemistry is controlled by both physical and biogeochemical processes. There are well-understood first-order principles that relate to ocean temperature and salinity to upper ocean carbon; using these relationships, the observations used in previous heat content mappings may be exploited in a similar manner for carbon mappings. Increases in temperature reduce the uptake of $CO_2$ through gas solubility (Weiss, 1974). Increases in salinity allow for further $CO_2$ uptake by increasing alkalinity (Williams and Follows, 2011). Additionally, temperature and salinity provide constraints on ocean circulation, which alters the background vertical gradients of both heat and carbon (Thomas et al., 2018; Williams et al., 2021). Anthropogenic carbon uptake in the high latitudes is constrained by salinity and stratification, which can be taken to be proxies for water mass formation (Terhaar et al., 2020, 2021; Bourgeois et al., 2022). The relationships between temperature, salinity, and carbon are regionally dependent as ocean dynamics and biology can set different drivers of $CO_2$ uptake (Lauderdale et al., 2016). If these relationships can be exploited, the increase in ocean observations from the Argo programme may provide valuable information that can help reconstruct ocean carbon fields alongside temperature and salinity fields. While the ocean carbonate system can also be approximated using other observations such as pH and salinity, for this study we focus on the potential benefit of observations present within the Argo dataset.

In this study, we apply an Ensemble Optimal Interpolation approach to reconstruct modelled upper-ocean dissolved inorganic carbon from synthetic temperature and salinity observations. Covariance fields are constructed using an ensemble of 6 Earth system models from the Climate Model Intercomparison Project Phase 6 (CMIP6). In this proof-of-concept study we show the potential skill available in using model covariance fields and Argo-style synthetic measurements to reconstruct carbon content between 0-2000m. Synthetic reconstructions of modelled ocean carbon are created by using pseudo-observations of temperature and salinity, similar to the ocean heat content synthetic reconstructions in Allison et al. (2019). The errors within these reconstructions can then be compared with those from the climatological fields to see where and how the method best works in the model world.

The work in this study is set out as follows. Section 2 introduces the Ensemble Optimal Interpolation scheme for ocean DIC using an ensemble of 6 CMIP6 models and the experiments used to test the reconstruction skill. The method is first assessed by reconstructing DIC inventory changes over the top 100m, and then over the top 2000m. Section 3.1 presents the ensemble correlation fields between DIC over the top 100m and temperature and salinity at the same location, with a discussion as to how these correlations relate to physical controls on the ocean carbon response. In Sections 3.3 and 3.4 the reconstruction potential of different temperature and salinity sampling distributions is assessed for the upper 100m, ranging from perfect spatial coverage to coverage more representative of Argo observations. The method is then extended to consider the full 2000m profiled by Argo floats in Section 4, and the potential to reconstruct recent global carbon changes is illustrated with a reconstruction using time-varying synthetic Argo observations in Section 5. Lastly, in Section 6 we discuss the potential and shortcomings of this method, both in terms of the setup with temperature and salinity and how the method may be expanded with other oceanographic observations.

## 2 The Ensemble Optimal Interpolation method for ocean carbon

Optimal interpolation is a non-dynamical mapping approach that uses weights to propagate information from observations to regions without observations (Daley, 1991; Smith and Murphy, 2007). The optimal interpolation method involves creating an analysis $A$ at locations $i$ and times $t$ from the sum of a background state $B_i$ and a weighted sum of the difference between the observed and background values, also known as the observation increments. The observations can be the same property as the final analysis, or can be of other properties, i.e. temperature and salinity observations can be used to reconstruct DIC if there is a physical link between them. The inclusion of other physically relevant data produces a *multivariate analysis*. For observations $O$ at locations $k$ within a sampling region $K$, the optimal interpolation method can be represented as

$$A_i = B_i + \sum_{k \in K} w_k (O_k - B_k). \tag{1}$$

The crux of the optimal interpolation problem is thus finding a suitable solution for the weights $w_k$. The optimal weights are those that minimise the expected analysis error at each gridpoint $i$, calculated as the root mean squared error (RMSE) between the analysis and the truth $T_i$:

$$RMSE(A_i, T_i) = \sqrt{\frac{1}{N} \sum_t (A_i(t) - T_i(t))^2}. \tag{2}$$

In this work we explore how the relationships between DIC and observed ocean variables such as temperature and salinity can be used to reconstruct upper ocean carbon inventories down to a depth of 2000m. The use of optimal interpolation to reconstruct ocean carbon from available carbon measurements is limited by the poor temporal and spatial coverage of existing DIC observations. However, temperature and salinity observations are more plentiful, particularly in the Argo era, so we explore the extent to which these observations could be used to reconstruct DIC. To avoid the problems associated with sparse input data, we take a multivariate analysis approach to reconstruct carbon from extensive synthetic observations of ocean temperature $T$ and salinity $S$ consistent with observations from the Argo programme (Wong et al., 2020), as well as annual average atmospheric $CO_2$ concentrations. On annual and longer timescales, atmospheric $CO_2$ is well-mixed, and so the use of annual atmospheric $CO_2$ measurements can allow the analysis to capture longer-term DIC changes from changes in the global carbon budget. With background fields for DIC, T, and S taken to be their global climatologies (i.e., $DIC(i,t) = \overline{DIC}(i) + DIC'(i,t)$, where $\overline{DIC}(i)$ is the time-average DIC concentration at location $i$ over the period 1955-2014), the optimal interpolation scheme is formulated to calculate the residual $DIC'$ from $T', S'$ within a region $K$:

$$DIC'(i,t) = \sum_{k \in K} (w_{C,k} \mathrm{pCO}_2'(t) + w_{T,k} T'(k,t) + w_{S,k} S'(k,t)), \tag{3}$$

where $w_{C,k}$ is the local weighting for the annual global averaged atmospheric $pCO_2$ and $w_{T,k}$ and $w_{S,k}$ are the weightings for observed ocean temperatures and salinities within the sampling region $K$. These optimal weights describe how information is propagated from atmospheric $pCO_2$ and hydrographic observations to the ocean carbon system, taking into account the interdependencies between input variables.

## 2.1 Choices for calculating background error covariances

The optimal weights $w_k$ in (1) and (3) are determined by the covariances between the background errors $B_i - T_i$. The covariance fields describe how information should be propagated from areas with observations to those without; the optimal solution will consider how much new information observations provide to the reconstruction and will depend on local and larger-scale relationships within the climate system. Any ensemble optimal interpolation method requires decisions on how the background error covariances are calculated, and which covariances are included to solve for the weights $w_k$. In the following we describe how we have made these decisions using a CMIP6 multi-model ensemble and with various assumptions as to which observations are used for the multivariate DIC analyses.

### 2.1.1 Background covariances from a CMIP6 ensemble

Climate model outputs can be used to provide covariance fields for optimal interpolation solutions. By providing complete pseudo-data in both space and time, climate models avoid some of the errors that arise from the coarseness of observational or parameterised covariance fields. Background covariance fields from global climate models have been used to reconstruct ocean temperatures and salinities from observations as well as to initialise decadal forecasts (Smith and Murphy, 2007; Smith et al., 2015; Cheng and Zhu, 2016). However, the model ensemble background covariance field will still contain errors that will need to be evaluated using sensitivity testing before the fields can be used with real-world observations.

To construct the background covariance fields, we obtained ocean potential temperature, ocean salinity, and ocean DIC model output from 6 CMIP6 Earth System models with a nominal horizontal resolution around $1°$ (Table 1). The output was taken from the historical experiments and covers the period of year 1955 to year 2014. This period was chosen as it has consistent behaviour in atmospheric $CO_2$ concentrations and is long enough to allow some multidecadal variability to be captured in the covariance fields. For each model, 5 realisations were used to capture the models' internal variability.

**Table 1.** CMIP6 Earth System Models and realisations used for reconstruction

| Model (Reference) | Realisations |
| --- | --- |
| ACCESS-ESM-1.5 (Ziehn et al., 2020) | r1i1p1f1, r2i1p1f1, r4i1p1f1, r5i1p1f1, r6i1p1f1 |
| CanESM5 (Swart et al., 2019) | r10i1p1f1, r11i1p1f1, r12i1p1f1, r13i1p1f1, r14i1p1f1 |
| CESM2 (Danabasoglu et al., 2020) | r1i1p1f1, r2i1p1f1, r3i1p1f1, r4i1p1f1, r5i1p1f1 |
| IPSL-CM6A-LR (Boucher et al., 2020) | r1i1p1f1, r2i1p1f1, r3i1p1f1, r4i1p1f1, r32i1p1f1 |
| MPI-ESM1.2-LR (Mauritsen et al., 2019) | r1i1p1f1, r2i1p1f1, r3i1p1f1, r4i1p1f1, r5i1p1f1 |
| UKESM1 (Sellar et al., 2019) | r1i1p1f2, r2i1p1f2, r3i1p1f2, r4i1p1f2, r8i1p1f2 |

For all variables, annual averages were calculated from monthly mean outputs, and outputs were bilinearly regridded from their native horizontal grids to a $1° \times 1°$ grid using the Python package `xESMF` (https://doi.org/10.5281/zenodo.1134365). The drifts in temperature, salinity, and DIC were calculated and removed by subtracting linear trends at each ocean grid cell in the

piControl runs. Oceanic variables were further integrated vertically in layers that span the ocean surface to 2000m depth. The integrated layers were chosen to be 0m-100m, 100m-500m, 500m-1000m, and 1000-2000m.

## 2.1.2 Spatial limits on background covariances

The sampling region $K$ for temperature and salinity observations in (3) can vary from co-located observations to global observations; the choice of sampling region requires balancing the extra information provided by additional observations with spurious propagation of information through errors in the covariance fields. In this study we use two idealised sampling methods to explore the limits of this first-order reconstruction of ocean carbon:

1. The first reconstruction method assumes full global coverage of temperature and salinity observations. For these synthetic reconstructions, the ocean inputs to reconstruct ocean DIC are co-located model temperature and salinity anomalies, as well as globally-uniform atmospheric pCO$_2$ anomalies. The resulting system has 3 input parameters to solve for ocean carbon at each grid cell. With this method, we explore how the local relationships between upper-ocean DIC, temperature, and salinity can be used to reconstruct carbon with perfect hydrographic knowledge. Since observations are complete in
this test, the presence of errors can be attributed to poorly known relationships or nonlinear dynamics within the model ensemble.

2. Synthetic observations are taken irregularly, consistent with the distribution of Argo profiles (Wong et al., 2020). While coverage of ocean temperature and salinity observations is significantly higher than those for ocean carbon, sampling remains irregular in space and time, so real-world reconstruction methods must account for this irregularity. We take
Argo data from years 2002-2015 and bin the profiles onto the $1° \times 1°$ horizontal grid. Any grid cell that has Argo observations for at least 6 months within a given year (running from January to December) is taken to be sufficiently observed, and the modelled annual average profile there is used as an observation for the reconstruction. For year 2002, this leads to a distribution of synthetic profiles that is heavily concentrated in the Northern Hemisphere and the North Atlantic in particular (Fig. 1a). By year 2015, the scheme allows for sampling of most of the global ocean outside the
Southern Ocean and the Arctic Ocean (Fig. 1b). Using this distribution of available profiles, we create reconstructions using time-varying profile locations and test how profile density impacts the reconstruction by setting the distribution as constant in time.

As the Argo profiles have incomplete global coverage, we conduct additional sensitivity tests to see how nearby Argo observations can be used to construct near-global reconstructions. For these reconstructions, the reconstruction at any given
point is made to be a linear combination of observations within a certain radius. Radii of $1°$, $2°$, or $5°$ are chosen: for a radius of $5°$ and current Argo coverage like that for year 2015, most regions of the ocean have observations within the search radius, allowing for nearly globally complete DIC reconstructions.

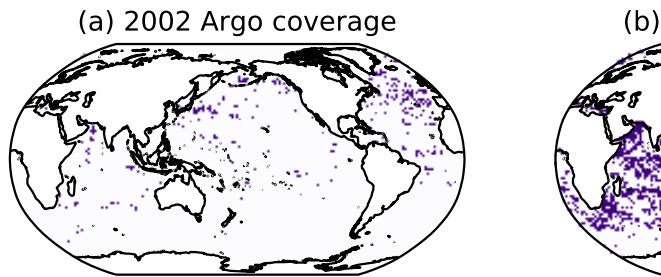

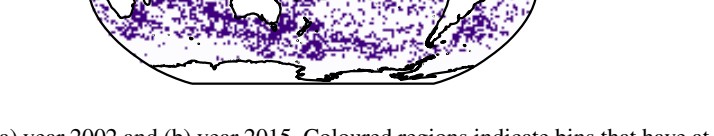

**Figure 1.** Argo profile distributions consistent with sampling for (a) year 2002 and (b) year 2015. Coloured regions indicate bins that have at least 6 months of observations in the given year.

## 2.2 Synthetic reconstructions and tests

Evaluating how well an analysis reproduces variability requires a comparison with a field truth. As the ocean carbon field is not known to a high accuracy, synthetic reconstructions of models' ocean carbon fields are created using various distributions of modelled temperature and salinity (Allison et al., 2019). For these tests, the synthetic observations come from the models and therefore contain no errors outside of small errors possible from sampling and regridding. As the synthetic observations can be assumed to have negligible error, the weights $w_k$ in (2) can be solved for through a least-squares algorithm, without having to consider observational errors (Smith and Murphy, 2007).

The models used in the synthetic reconstructions are the same models used to calculate the ensemble covariance fields. The inclusion of a model in the ensemble can over-fit the results and produce spuriously accurate reconstructions; therefore, for a synthetic reconstruction of, for instance, UKESM1 DIC inventories, we eliminate all the UKESM1 realisations from the ensemble. The covariances and optimal weights are then calculated from the remaining ensemble members. Then, these optimal weights are used with the UKESM1 temperature and salinity profiles to create a DIC analysis.

To compare the reconstructions, the improvement in the RMSE is calculated relative to the RMSE of the first-guess background using:

$$\varepsilon_i(A) = \frac{RMSE(B_i, T_i) - RMSE(A_i, T_i)}{RMSE(B_i, T_i)}. \tag{4}$$

Here the conventions follow those in (1), where $RMSE(B_i, T_i)$ is the RMSE between the modelled truth $T$ and the background climatology field $B$ (equivalent to the standard deviation of $T$), and $RMSE(A_i, T_i)$ is the RMSE between the modelled truth and the multivariate analysis $A$. Each RMSE is calculated for the same historical period used in the creation of the model ensemble, i.e. $t$ is taken from output within the modelled years 1955 to 2014. The maximum value of 1 indicates a perfect reconstruction $A$, whereas values below 0 indicate that the errors are larger for the analysis than they are if the solution were to be taken as the climatological first-guess.

For the following analysis, synthetic reconstructions of DIC are first made over the upper 100m, and then over the upper 2000m, using both global and irregular temperature and salinity observations. Over the top 100m, the ocean carbon distribution is expected to reflect the controls of solubility and alkalinity on ocean carbon, due to the dominance of air-sea gas exchange on

the carbon inventory. Over the upper 2000m, ocean circulation and ventilation are expected to play a larger role in determining the reconstructed carbon fields.

## 3 Reconstruction of ocean DIC within the top 100m

Our reconstruction approach using the ensemble of 6 CMIP6 model runs is now applied to carbon inventory changes in the top 100m. We begin by exploring the covariance fields between the system input and output variables through correlations. As the covariance fields remain the same for both synthetic and real-world analyses, it is important to test whether these fields reflect the physical relationships between DIC, $pCO_2$, T, and S.

### 3.1 Model correlation fields between $pCO_2$, temperature, salinity, and DIC

The least-squares solution for the weights for temperature, salinity, and atmospheric $CO_2$ is a function of the covariances between the input variables and DIC. Thus, the structure of the covariance fields provides insight into how temperature and salinity can be used to reconstruct carbon. For simplicity, we illustrate these relationships through correlation fields taken for the entire model ensemble, which normalise the relationships using the variances of each input variable.

#### 3.1.1 Correlations with atmospheric $pCO_2$

The correlation fields between atmospheric $pCO_2$ and DIC, temperature, or salinity reflect the impact of emissions on the ocean mixed layer. Atmospheric $pCO_2$ is strongly positively correlated with mixed layer DIC and reaches values near 1 in the mid-latitude ocean (Fig. 2a). This strong correlation reflects how the ocean takes up carbon under higher atmospheric $pCO_2$ due to gas disequilibrium. The correlation between atmospheric $pCO_2$ and integrated temperature changes is widely positive, consistent with the long-term ocean uptake of both heat and carbon during carbon emissions (Fig. 2b). Lastly, the correlation between atmospheric $pCO_2$ and integrated salinity changes is close to zero in most regions, although smaller regions such as the Gulf Stream show stronger correlations.

#### 3.1.2 Correlations between DIC and temperature

Whereas atmospheric $pCO_2$ has little interannual variability, mixed layer temperature, salinity, and DIC will have variability on interannual and decadal timescales alongside long-term trends. The correlation fields between mixed layer DIC, temperature, and salinity reflect the combined response to external forcing from atmospheric $pCO_2$ and internal variability.

Mixed layer temperature and DIC are positively correlated in most of the ocean, with strong negative correlations in the eastern equatorial Pacific and Indian Oceans (Fig. 2d). This positive correlation contradicts the negative correlation expected from the solubility control of temperature on DIC.

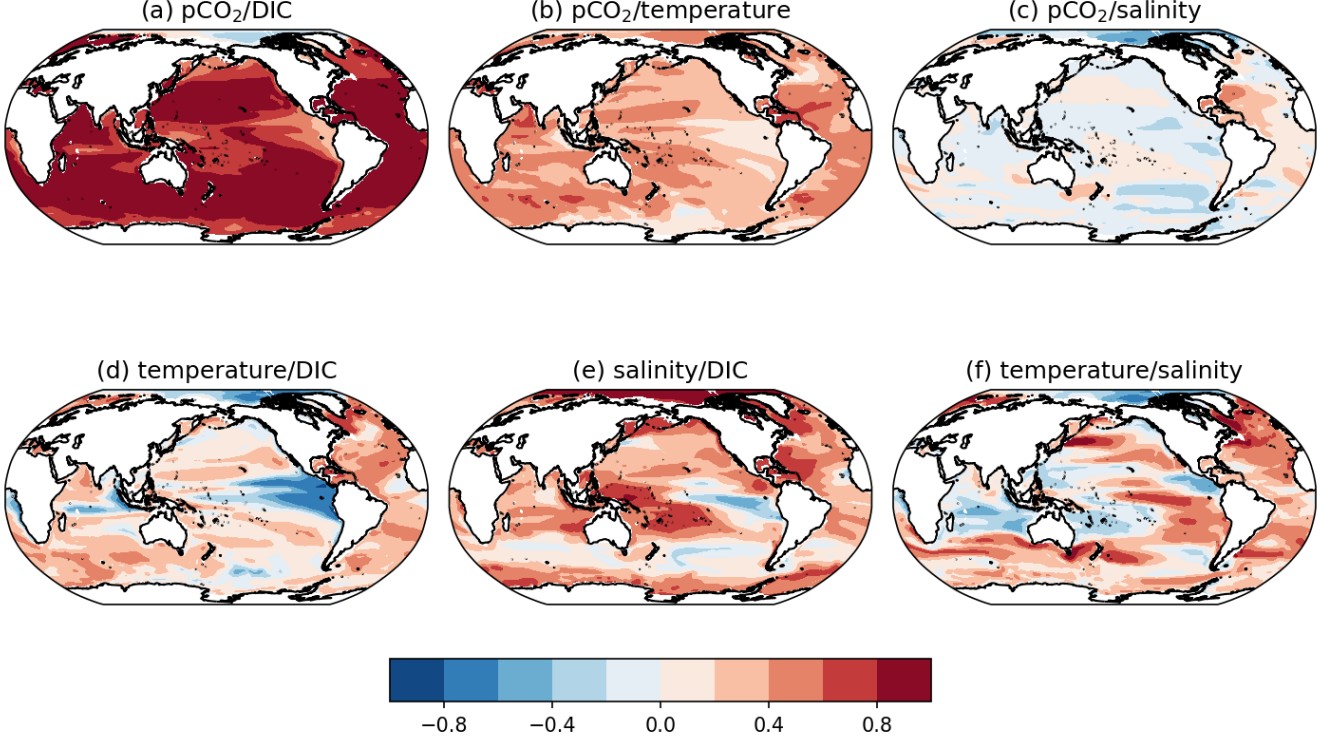

**Figure 2.** Correlations for the entire CMIP6 historical ensemble between changes in (a) atmospheric $pCO_2$ and upper-ocean (0-100m integrated) DIC, (b) atmospheric $pCO_2$ and upper-ocean temperature, (c) atmospheric $pCO_2$ and upper-ocean salinity, (c) upper-ocean temperature and DIC, (e) upper-ocean salinity and DIC, and (f) upper-ocean temperature and salinity.

To separate the correlation into terms relating to the external forcing and ocean variability, we can decompose temperature

and DIC into terms proportional to atmospheric $pCO_2$ and an anomaly term:

$$T'(x,y,t) = \alpha(x,y)pCO_2(t) + T_a(x,y,t) \tag{5}$$

$$DIC'(x,y,t) = \gamma(x,y)pCO_2(t) + DIC_a(x,y,t) \tag{6}$$

The terms $\alpha, \gamma$ describe the respective spatial patterns of changes in $T', DIC'$ with changes in $pCO_2$. The anomalies of $T', DIC'$ after a regression against $pCO_2$ is removed are denoted as $T_a$ and $DIC_a$.

With the decomposition in (5) and (6), the correlation between DIC' and T' can be approximated as the sum of a $pCO_2$ and a non-$pCO_2$ component, where $\sigma_{DIC}, \sigma_T$ are the standard deviations of $DIC'$ and $T'$:

$$\rho(DIC',T') \approx \underbrace{\alpha\gamma\frac{\mathrm{var}(pCO_2)}{\sigma_{DIC}\sigma_T}}_{pCO_2 \text{ term}} + \underbrace{\frac{\mathrm{cov}(DIC_a,T_a)}{\sigma_{DIC}\sigma_T}}_{\text{non-}pCO_2 \text{ term}}. \tag{7}$$

This decomposition is accurate to first order. A more detailed derivation of this breakdown and a discussion of the approximation can be found in Appendix A.

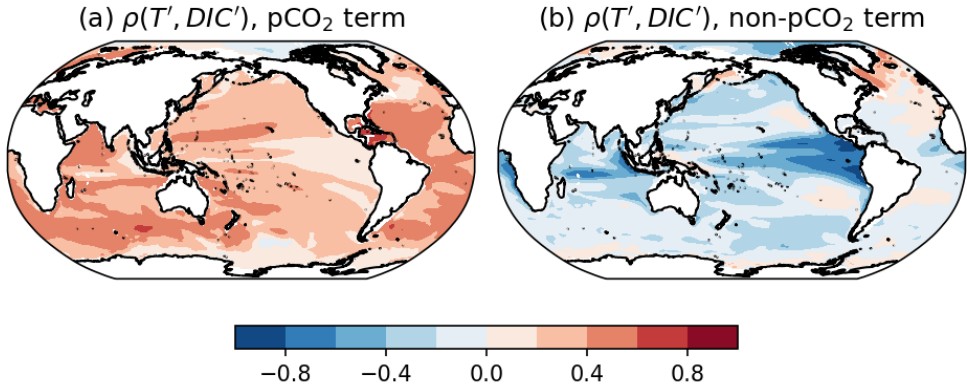

**Figure 3.** Breakdown of the correlation between upper-ocean DIC and temperature, $\rho(DIC', T')$, from Fig. 2d: (a) a term proportional to the variance of atmospheric $pCO_2$, and (b) a term consisting of the covariance of the residuals calculated after removing a linear fit against atmospheric $pCO_2$. For discussion of the breakdown of the correlation into terms and on the approximation, see Appendix A.

The $pCO_2$ component of the correlation field between temperature and DIC is positive almost everywhere (Fig. 3a). This relationship is consistent with the combined oceanic uptake of heat and carbon from $CO_2$ emissions. Conversely, the anomaly term for the correlation field between temperature and DIC is broadly negative, which corresponds with the solubility control for DIC (Fig. 3b). Thus the heterogenity found in the overall correlation between DIC and temperature in Figure 2 can be understood as the sum of an emissions-driven undersaturated response that correlates to but is not driven by warming, and a

temperature-driven solubility response.

### 3.1.3   Correlations between salinity and DIC or temperature

Unlike DIC, temperature, and atmospheric $pCO_2$, mixed-layer salinity changes have weak trends. Decomposing the correlations between salinity and temperature or DIC show little role for the $pCO_2$ term in (7) (Appendix A). Therefore we continue by exploring the full correlation fields with mixed layer salinity.

The correlations between upper-ocean salinity and upper-ocean DIC are positive in most regions, reflecting the alkalinity control on carbon solubility (Fig. 2e). Correlations are weak and slightly negative in the Southern Ocean and eastern equatorial Pacific, respectively. These correlations may be imprints of dynamical changes; for the Southern Ocean, the salinity in the frontal zone impacts the strength of mode and intermediate water formation (Terhaar et al., 2022), while for the equatorial Pacific enhanced wind-driven upwelling transports fresher, carbon-rich waters to the surface (Williams et al., 2021). Lastly,

correlations between upper-ocean temperature and salinity are moderate and show strong regional variability (Fig. 2f).

     Overall, the correlation fields show first-principle controls relating to the increase of ocean heat and carbon under emissions, as well as solubility and alkalinity controls of temperature and salinity anomalies on ocean DIC.

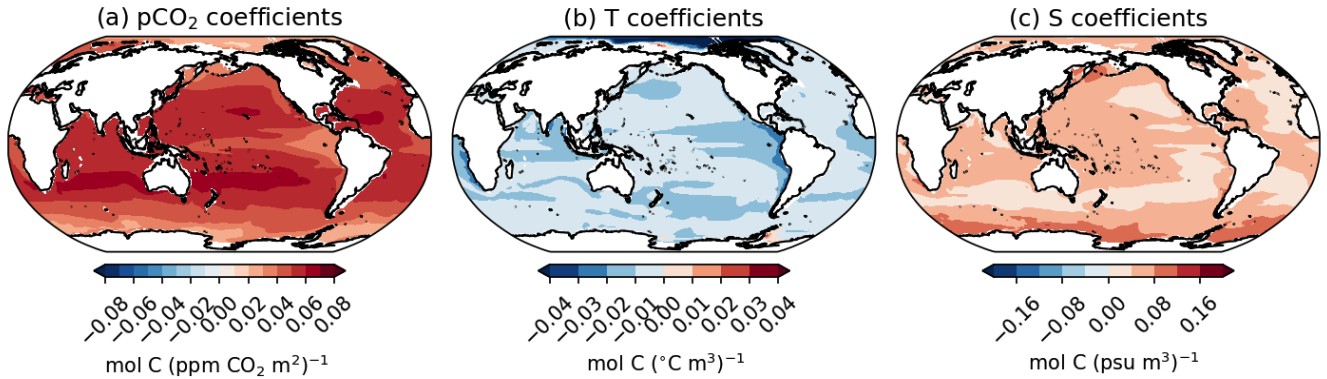

**Figure 4.** Ensemble coefficients for reconstructing upper-ocean (0-100m integrated) DIC with co-located observations. (a) Coefficients for annual average atmospheric $CO_2$ concentrations, in units mol C (ppm $CO_2$ m$^2$)$^{-1}$, (b) coefficients for upper-ocean integrated temperature, in units mol C ($^\circ$C m$^3$)$^{-1}$, and (c) coefficients for upper-ocean integrated salinity, in units mol C (psu m$^3$)$^{-1}$.

## 3.2 Ensemble optimal weights for pCO$_2$, temperature, and salinity

Translating covariance and correlation fields to optimal mapping parameters involves both the relationships between the in-
put variables and the output variables as well as the relationships between input variables. Therefore, optimal solutions are nontrivial; for instance, regions with similar correlations between DIC and temperature and salinity (such as the equatorial Pacific) may have coefficients with different signs depending on how the system fits the linear model to the data. We continue by comparing the least square coefficients fit to atmospheric $pCO_2$, integrated ocean temperature, and integrated ocean salinity to the correlations in Section 3.1 and the first-principle controls on carbon storage.

Increases in atmospheric $pCO_2$ are translated to increases in upper-ocean carbon (Fig. 4a). The weight magnitudes reach local maxima in the subtropics, in accordance with regional variability in the Revelle buffer factor, which describes the ratio between increases in DIC and increases in atmospheric $pCO_2$ (Williams and Follows, 2011) and thermocline ventilation. Temperature coefficients are negative almost everywhere (Fig. 4b), while salinity coefficients are positive everywhere (Fig. 4c). While the correlations between upper-ocean DIC, temperature, and salinity show regional variability, the ensemble ultimately
reveals a consistent forced control of atmospheric $CO_2$, solubility control by temperature, and alkalinity control by salinity.

Thus, the least-squares solution for DIC as a combination of atmospheric $pCO_2$, temperature, and salinity changes is able to capture the controls of the Revelle buffer factor, solubility, and alkalinity on upper-ocean carbon. The information from ocean temperature and salinity observations are complementary: regions with relatively high/low temperature coefficients correspond with low/high salinity coefficients. The optimal weight solutions indicate that the information provided by these observations
are consistent with our hypothesised first-order controls.

## 3.3 Reconstruction potential using co-located observations

Sensitivity tests are now conducted to estimate the ability to reconstruct ocean carbon, as well as explore how the ensemble composition impacts the reconstruction. In total, 6 reconstructions were created, in which one model was removed from the ensemble, the covariance fields between $DIC', T', S'$, and atmospheric pCO$_2$ are recalculated using the remaining models, and then $DIC'$ from the excluded model is reconstructed using the new covariance fields and its own $T'$ and $S'$ observations (Section 2). The coefficients for atmospheric CO$_2$, temperature, and salinity are similar across the sensitivity setups with each model removed (Supplementary Figures S1-S3), so we continue by comparing the RMSE improvements $\varepsilon$ from (4) for each sensitivity test.

The ensemble minimum, average, and maximum relative RMSE improvements are calculated for 6 reconstructions, each created by eliminating a model from the CMIP6 ensemble. The minimum and maximum error improvements $min(\varepsilon)$ and $max(\varepsilon)$ reflect the makeup of the ensemble. A negative minimum error improvement at any point indicates where one of the models in the ensemble has covariances that are substantially different than the others; therefore, including the model within the ensemble adds uncertainty to the reconstruction and pushes the reconstruction towards the first-guess climatology field. A high maximum error improvement indicates that the solution weights for the sensitivity results are similar. This similarity can arise from strong physical constraints on the upper ocean carbon system but may also arise because the ensemble members are spuriously similar in their architecture or representations of climate processes.

The RMSE improvement of the reconstruction relative to the climatological first-guess is now considered for both the full upper-ocean carbon response and the detrended carbon response (Fig. 5). Over multiple decades, we expect the carbon inventory response to be dominated by an upwards trend due to continuing carbon emissions; thus, the improvement in the detrended response provides insight into how interannual and decadal variability in ocean carbon storage is reproduced. Areas where the RMSE increases relative to the climatological first guess are noted in red. For this analysis we focus on the open-ocean RMSE reductions, as the CMIP6 models have different coastlines after being regridded, leading to small areas of RMSE increases near the land/sea boundary.

Outside of some coastal and Arctic regions, each sensitivity test reduces the upper-ocean RMSE for upper-ocean carbon (Fig. 5a). This improvement is not due solely to the reproduction of the long-term increase in ocean carbon, as most regions show improvements in detrended ocean carbon as well; exceptions to this improvement can be found in the high latitudes and subtropical regions (Fig. 5d). The average reconstruction reduces the RMSE by between 60% to 90% on average, relative to the climatology first-guess RMSE (Fig. 5b) and reduces the detrended RMSE by 30% to 80% (Fig. 5e). These relative improvements are equivalent to the method capturing over 60% of the detrended variability in most regions of the ocean. The high (close to 1) relative RMSE reductions found in the ensemble maximum statistics (Figs. 5c and e) suggest that there are models which are highly similar to one another.

For reconstructions of global DIC inventories using co-located temperature and salinity observations, the sensitivity experiments show an average RMSE reduction of 93%. When considering detrended DIC inventories, the sensitivity experiments reduce the RMSE by 68% on average.

There are noticeable regional variations throughout all of the improvement statistics. The western low-latitude Pacific and subtropical Indian ocean show consistent local maxima in all of the improvement statistics. These regions of maximum relative RMSE reduction are characterised by their strong correlation between salinity and DIC, suggesting that the most constrained responses within the CMIP6 ensemble may be related to the control of alkalinity on DIC. Conversely, the small regions that show potential degradation in the reconstructions due to the errors in the covariances are characterised by weak correlations against atmospheric $pCO_2$ changes (for the full carbon signal reconstruction, Fig. 2a-c) and weakly positive covariances between $T_a$ and $DIC_a$ (for the detrended carbon system, Fig. 3b).

Thus, when considering pointwise observations of temperature and salinity, alongside global average $CO_2$ concentrations, our sensitivity experiments indicate that a substantial amount of upper-ocean carbon variability can be reconstructed. These estimates can provide an upper bound on the reconstruction potential as global coverage of temperature and salinity observations is theoretically ideal but difficult to accomplish, even with widespread autonomous observing tools. We therefore continue by exploring how irregular observations of temperature and salinity impact reconstructions of upper-ocean carbon.

### 3.4 Reconstructing carbon using irregular Argo observations

The pointwise reconstructions of upper-ocean carbon from atmospheric $pCO_2$, temperature, and salinity reveal that the physical controls of solubility and alkalinity can explain a substantial amount of interannual variability in the upper-ocean carbon system. Although temperature and salinity profiles have a larger global coverage than ocean carbon profiles, coverage remains irregular and incomplete. In this section we explore how the irregular coverage in Argo profiles impacts carbon reconstructions by conducting further synthetic reconstructions with observations consistent with year 2015 Argo coverage (Fig. 1b).

When considering irregular sampling, reconstructions with global coverage use background covariances to propagate information from Argo profile locations to the rest of the ocean. There is the potential for information from multiple Argo sites to be used to reconstruct ocean carbon at a given location. If the covariance fields are correct, additional information should improve the reconstruction and increase the relative error reduction; however, any errors in the covariance fields can propagate and decrease the relative error reduction. The radius of influence for an observation should ideally balance the extra information and errors it provides in the reconstruction.

With the smaller radii of $1°$ and $2°$, the method creates an incomplete reconstruction even under current-day Argo profile distributions. With a larger radius of $5°$, most of the global ocean has nearby temperature and salinity profiles that can be used for the reconstruction (see Fig. 6 coverage). Overall, the benefit of increased coverage outweighs the increases in reconstruction errors (Supplementary Figure S4). Thus, in this section we focus on the $5°$ radius results for simplicity.

With the $5°$ cutoff radius for observations, the carbon reconstruction at each point will use between 0 and 81 (= 1 local observation + 80 observations within the cutoff radius) temperature and salinity inputs. Including the globally averaged atmospheric $CO_2$ inputs, DIC at each point can be reconstructed from at least 1 and up to 82 input variables (up to 81 temperature and salinity profiles + globally averaged atmospheric $pCO_2$).

Errors in spatial covariances result in some regions with lower RMSE improvements compared to reconstructions using co-located observations (Fig. 6a, d). Across the sensitivity tests and outside of the Southern Ocean and small regions in the

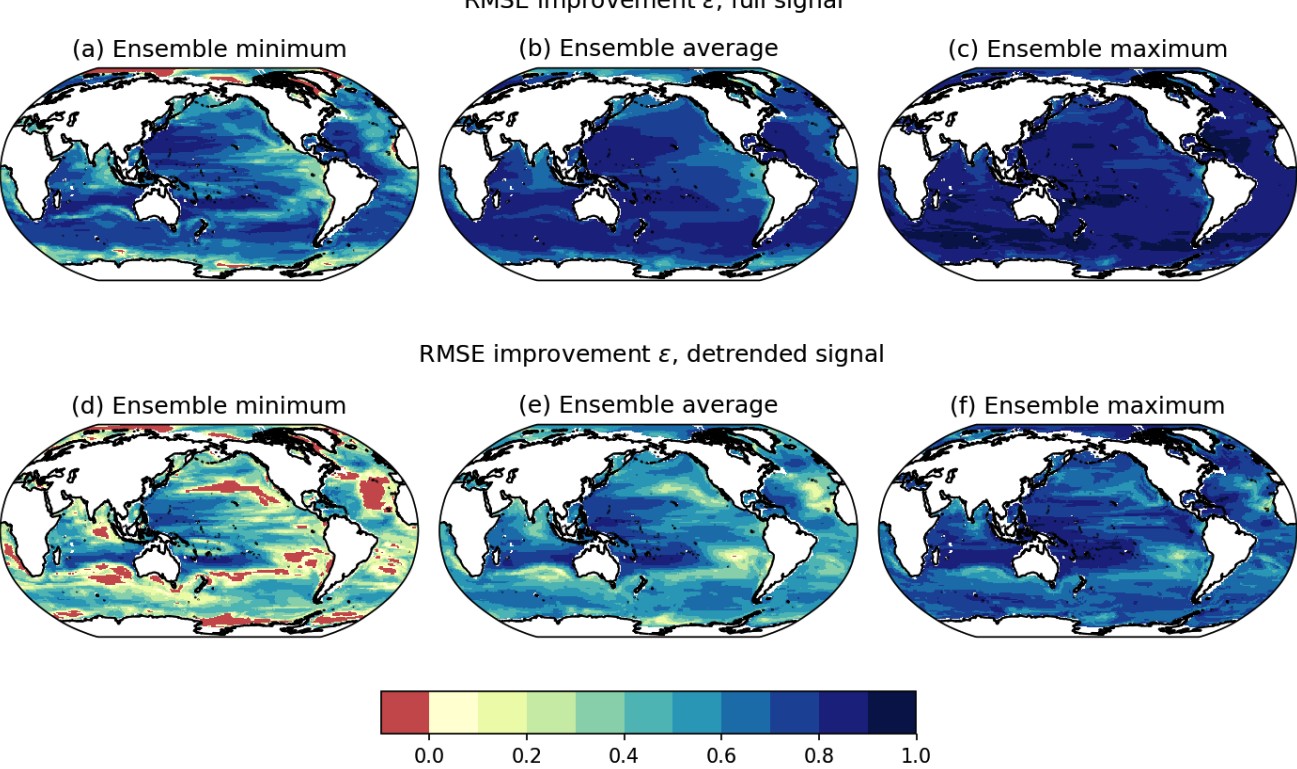

**Figure 5.** Ensemble statistics for the relative RMSE reductions $\varepsilon$ using co-located temperature and salinity over the upper 100m: (a) ensemble minimum, (b) ensemble average, and (c) ensemble maximum. Ensemble statistics for the relative RMSE reductions, but considering only the detrended carbon signal: (d) ensemble minimum, (e) ensemble average, and (f) ensemble maximum. Red areas indicate regions where the sensitivity tests show a RMSE increase relative to the assumption of climatology.

North Atlantic, North Pacific, and equatorial Pacific, carbon can be reconstructed with a relative RMSE reduction of at least
50%. When considering the detrended carbon signal most regions show an average RMSE reduction of at least 40%. Noticably, the ensemble minimum RMSE improvement for the detrended signal shows substantial regions where the sensitivity test that excludes MPI-ESM1.2-LR produces a poor reconstruction. The MPI-ESM1.2-LR may be viewed as an outlier model that provides additional uncertainties not covered by the other models.

For most of the upper ocean the RMSE improvements are similar to those found in the reconstruction with co-located
temperature and salinity measurements, and the regional variation in the RMSE improvements is similar regardless of whether co-located or removed observations are used. The additional information provided by nearby measurements counteracts the error propagation inherent in the scheme, suggesting that the use of irregular Argo profiles provides sufficient information to reproduce carbon variability in the mixed layer.

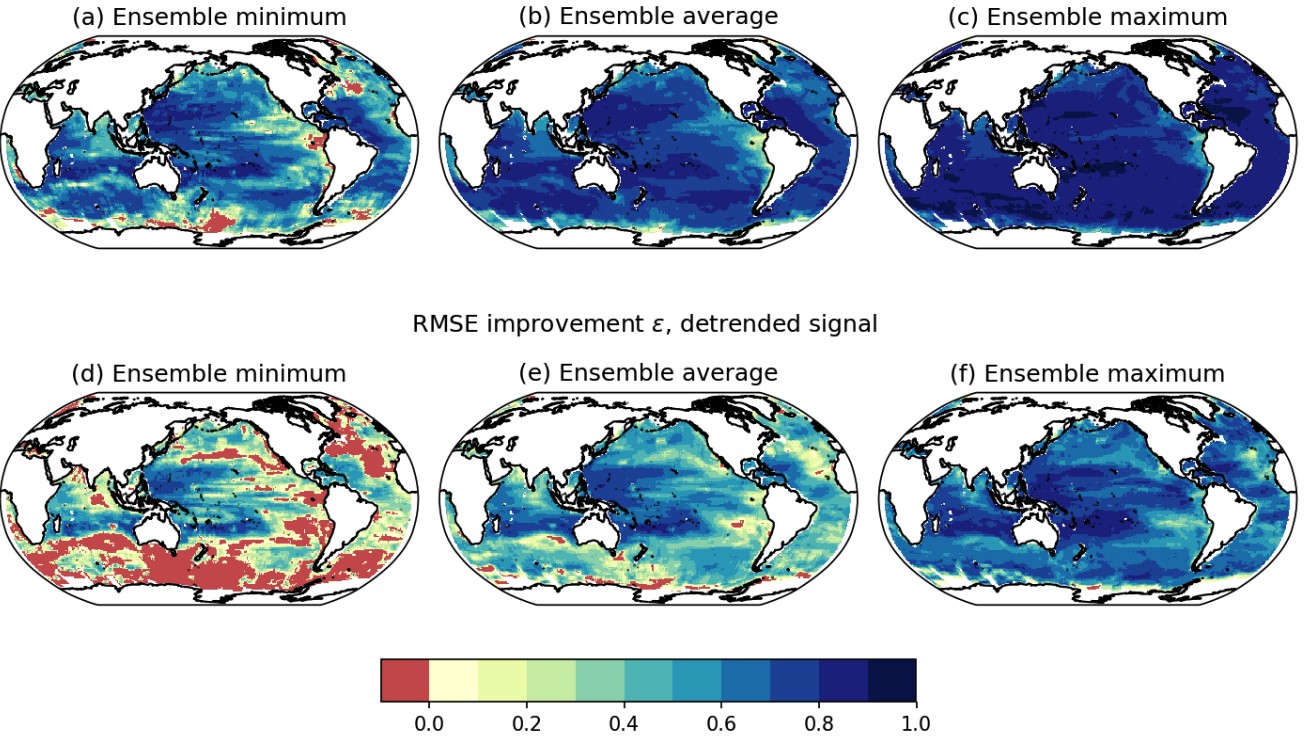

**Figure 6.** As for Fig. 5, but RMSE improvements over the upper 100m for the scheme that uses Argo 2015-type temperature and salinity observations within 5° of the reconstruction location.

## 4 Reconstruction of ocean DIC within the top 2000m

Our reconstruction approach is now applied to carbon inventories within the upper 2000m. Our assessment of the CMIP6 models indicate that the relationships between carbon, temperature, and salinity may be used to reconstruct ocean carbon variability within the near-surface ocean. Anthropogenic carbon uptake and decadal variability are concentrated in the top 1000m (De-Vries et al., 2017; Gruber et al., 2019a). As Argo profiles can provide temperature and salinity information down to 2000m, the synthetic reconstructions are extended to cover this vertical extent. For these interior reconstructions, the carbon inventories are reconstructed as a set of layers: 100m-500m, 500m-1000m, and 1000m-2000m. For the interior layers, we conduct the same analysis as was used for the top 100m reconstructions by first examining the uncertainties when reconstructing carbon using co-located temperature and salinity measurements, and then extending to reconstructions using irregular observations consistent with Argo profile locations.

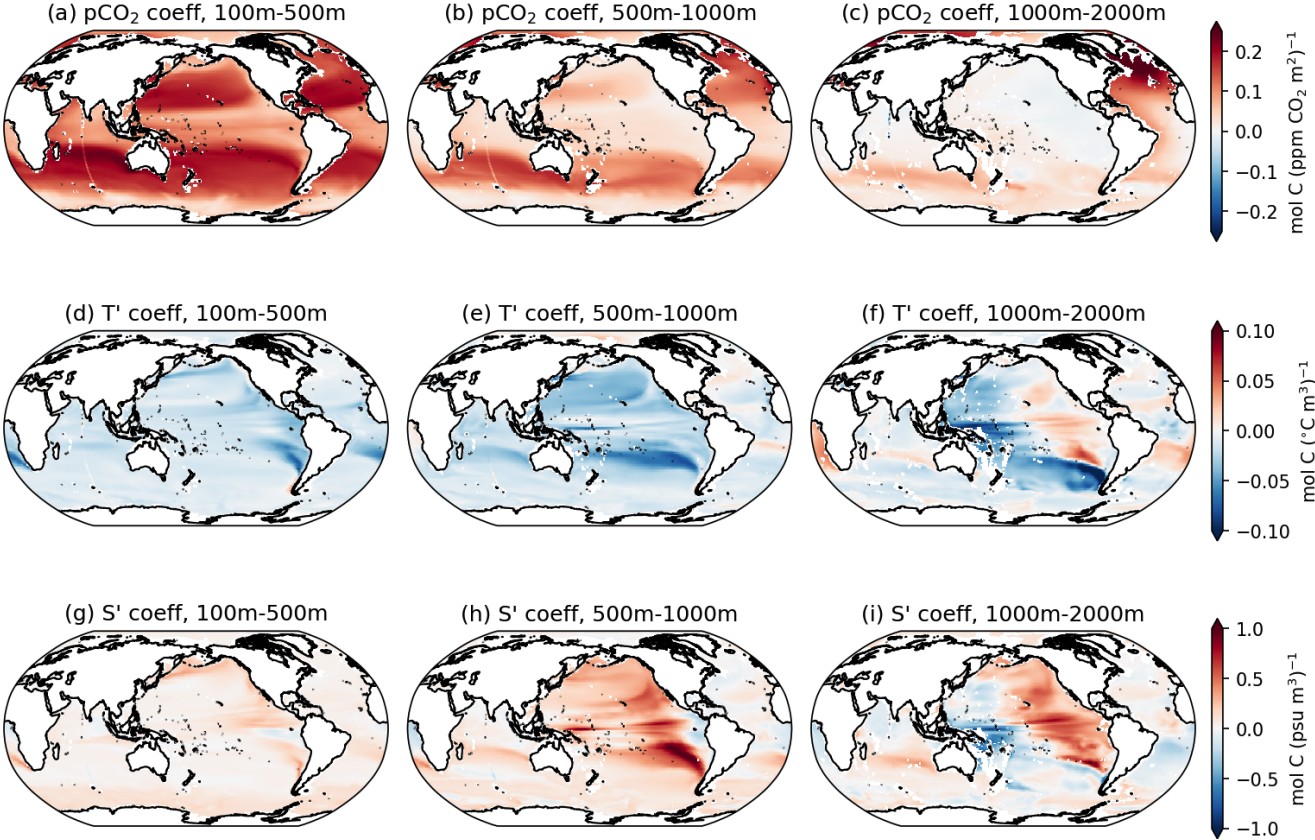

**Figure 7.** Optimal coefficients for interior DIC as a function of $pCO_2$ (in units mol C m$^{-2}$ ppm$^{-1}$, panels a-c), temperature (in units mol C m$^{-2}$ °C$^{-1}$, panels d-f), and salinity (in units mol C m$^{-2}$ psu$^{-1}$, panels g-i). Columns show coefficients for interior ocean layers: 100m-500m (a,d,g), 500m-1000m (b,e,h), and 1000m-2000m (c,f,i).

## 4.1 Sensitivity of optimal weights to depth level

The optimal coefficients for carbon as a function of co-located temperature and salinity, along with atmospheric $pCO_2$ concentrations, display distinct depth sensitivity (Figure 7). The optimal coefficients for atmospheric $pCO_2$ remain positive for most region and depth combinations. For the 100m-500m layer, $pCO_2$ coefficients reach their local maxima in the subtropical gyres, similarly to the optimal solution found for the upper 100m layer (Figure 4a). Below 500m, the optimal weights reflect regions of strong ventilation in the North Atlantic and Southern Oceans; below 1000m, the coefficients outside these regions are near
zero.

     The optimal coefficients for interior temperature changes remain negative for most regions above 1000m, with depth-dependent structure (Figure 7d-e). For the 1000m-2000m layer, the optimal temperature coefficients exhibit a zonal asymmetry in the Pacific basin, where the coefficients are negative in the western Pacific and positive in the eastern Pacific. Salinity

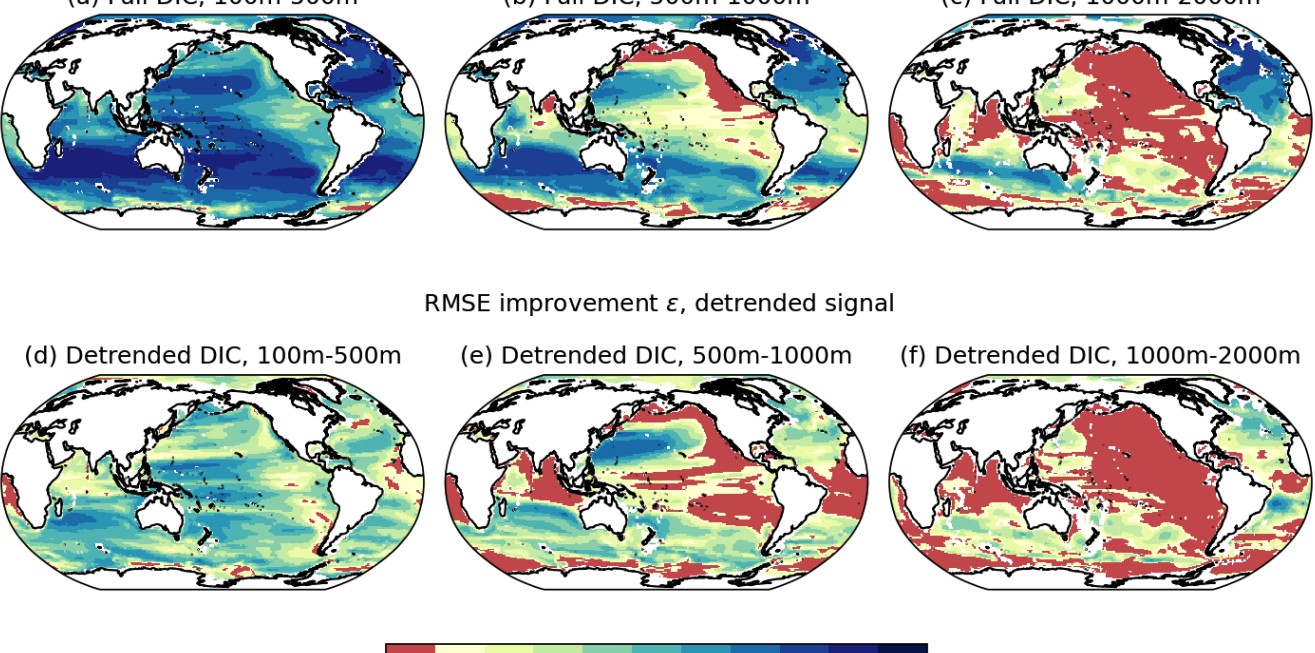

RMSE improvement $\varepsilon$, full signal

(a) Full DIC, 100m-500m  (b) Full DIC, 500m-1000m  (c) Full DIC, 1000m-2000m

RMSE improvement $\varepsilon$, detrended signal

(d) Detrended DIC, 100m-500m  (e) Detrended DIC, 500m-1000m  (f) Detrended DIC, 1000m-2000m

**Figure 8.** Average RMSE reduction $\varepsilon$, relative to a climatology, for sensitivity tests with out-of-sample models: (a-c) uses average $\varepsilon$ for the full DIC' signal and (d-f) uses average $\varepsilon$ for the detrended signal. The average RMSE reduction is calculated for individual depth levels: 100m-500m (a,d), 500m-1000m (b,e), and 1000m-2000m (c,f).

coefficients operate similarly to temperature coefficients; for the 100m-500m and 500m-1000m solutions salinity coefficients
are largely positive (Figure 7g-h). Below 1000m, salinity coefficients exhibit the same zonal structure seen in the temperature
coefficients. The optimal coefficients at this depth are spatially correlated, which suggests that the information provided by
temperature and salinities at this depth are less complementary than at shallower levels.

### 4.2  Carbon inventory reconstructions over 0-2000m using co-located observations

The skill of the Ensemble Optimal Interpolation reconstructions is now assessed for carbon inventory changes in the upper
2000m, following the prior assessment over the upper 100m.

For both the full carbon signal and the detrended carbon signal, the reconstruction skill decreases with depth. For carbon
within the top 100m of the water column, the average RMSE reduction is at least 60% in most regions, and for globally-

integrated DIC reconstructions the average RMSE reduction is over 90% (Figure 5b). This RMSE reduction for the full DIC signal drops to 30%-80% for the 100m-500m layer (Figure 8b)

Below 500m, the sensitivity tests show large regions where the reconstruction increases the RMSE relative to a climatological first guess: for 500m-1000m the eastern North Pacific and Southern Ocean show increased errors, whereas below 1000m most of the Pacific and Indian basins show increased errors (Figure 8c,d). The regions in which the reconstruction maintains skill with depth are the well-ventilated regions that show maximum optimal coefficients for $pCO_2$ (Figure 7c).

As with the reconstructions for the top 100m, reconstructions at depth also show a lower RMSE reduction for the detrended

DIC signal (Figure 7f-h). and 0%-60%. For carbon between 100m-500m, the reconstructin reduces the detrended RMSE by between 0-60% in most regions. Below 500m, the regions that exhibit increase errors in the detrended DIC signal are similar to those with larger errors in the full DIC signal.

While the sensitivity tests show poor reconstruction skill for large areas below 1000m, carbon changes below this depth horizon are generally small, and large negative relative errors are found in regions that have little variability (see Supplementary

Figure S5). Thus, the skill found in the 0-100m and 100-500m layers mitigates the impact of covariance errors in these deeper layers when considering full column reconstructions.

## 5    Potential of the method to provide global and regional carbon timeseries

To illustrate the potential for reconstructing upper-ocean carbon using temperature and salinity measurements, we reconstruct output from the Norwegian Earth System Model (NorESM2, Seland et al. (2020)) using the full ensemble covariance fields and

temperature and salinity measurements at locations similar to Argo observation locations. This reconstruction is an additional out-of-sample reconstruction, similar to those made for the error reduction statistics in Sections 3.3 and 3.4, but uses covariance fields constructed from the entire CMIP6 model ensemble. We elect to use NorESM2 as an independent check as it has 3 available realisations and uses isopycnal depth coordinates, whereas all the models in the CMIP6 ensemble are z-level models.

Within this reconstruction, the cutoff radii for our localization are depth-specific. The cutoff radii decrease with depth: 5°

for 0-100m, 2° for 100-500m, 1° for 500-1000m, and only using co-located observations for 1000-2000m. These cutoff radii were chosen to limit the influence of errors in the covariance fields on the final reconstruction.

The global upper-ocean carbon inventory is dominated by a positive, near-linear trend across all depth levels (solid lines, Figure 9a). Most carbon changes occur within the top 500m (blue and orange lines); at these levels the reconstruction captures the long-term global behavior. Below 500m, the errors in the covariance fields result in biases in the reconstruction. Between

500-1000m, the reconstruction overestimates DIC accumulation, whereas below 1000m the reconstruction underestimates DIC accumulation. We note that, particularly below 1000m, the enhanced coverage of the Argo program over time results in a less accurate reconstruction due to the errors in the ensemble covariance fields.

To show how the reconstructions operate on more regional scales, we provide reconstructed modelled carbon inventory at the Bermuda Atlantic Time Series (BATS, 31°50'N 64°10'W) and the Hawaii Ocean Time Series (HOT, 22°45'N 158°W) (Figure

9b,c). The multidecadal time series at BATS will be a useful validation tool for future reconstructions. With the localisation

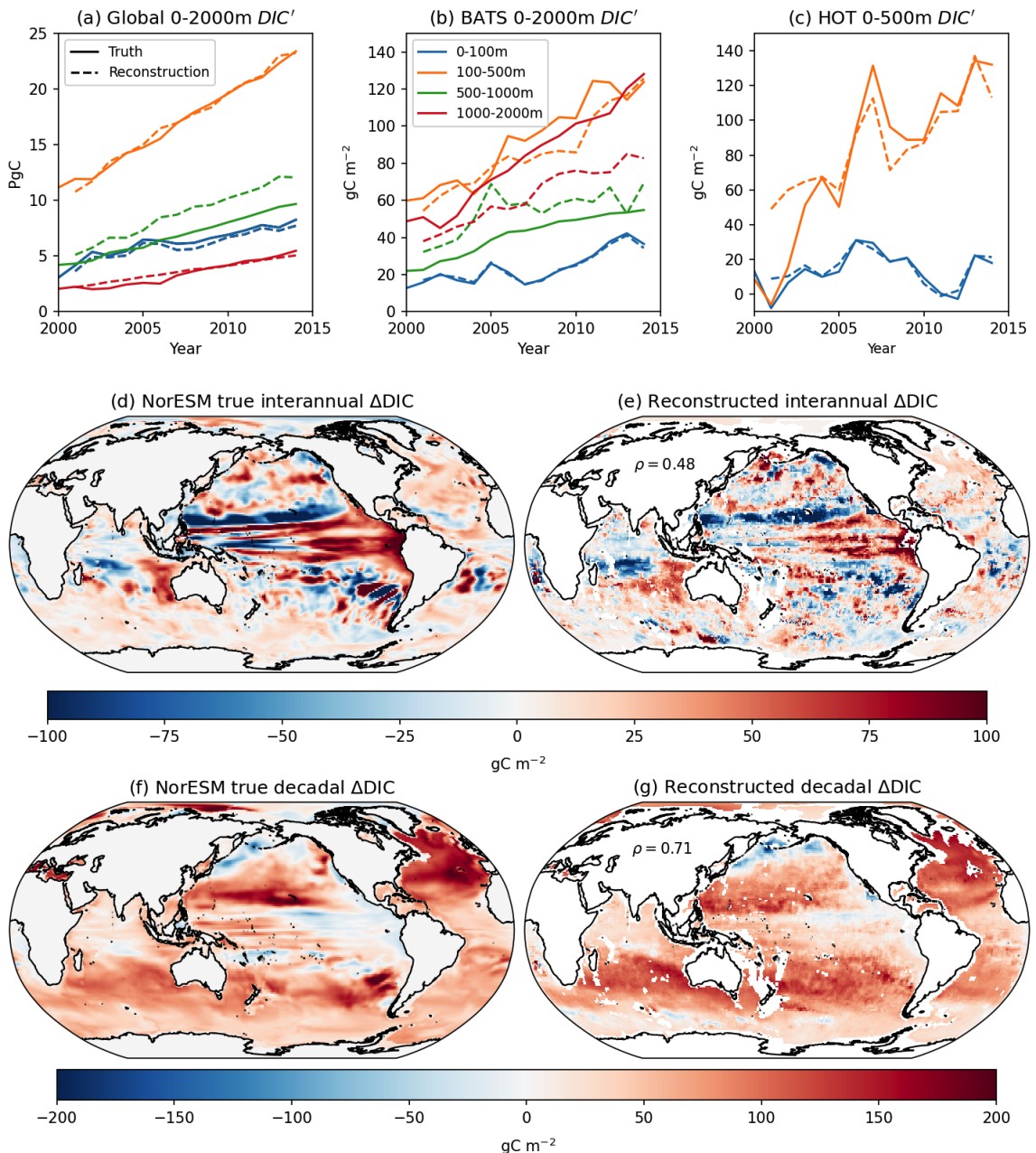

**Figure 9.** Reconstructions of NorESM 0-2000m carbon inventories. (a) Globally-integrated NorESM carbon inventories (solid lines) and reconstructed carbon inventories using temporally-varying Argo coverage (dashed lines), in units Pg C. Colours indicate depths: 0-100m (blue), 100-500m (orange), 500-1000m (green), and 1000-2000m (red). (b) As for (a), but vertically-integrated DIC concentrations at BATS, in gC m$^{-2}$. (c) As for (b), but vertically-integrated DIC concentrations at HOT to 500m, in gC m$^{-2}$. Snapshots of interannual changes in 0-2000m DIC, in units gC m$^{-2}$: (d) NorESM truth and (e) reconstruction. Interannual changes are taken as the difference between year 2013 and year 2014 carbon fields. Snapshot of near-decadal changes in 0-2000m DIC between periods 2001-2006 and 2009-2014, in units gC m$^{-2}$: (e) NorESM truth and (f) reconstruction.

procedure, BATS has observations for a reconstruction for the top 2000m; at HOT a lack of observations restricts the analysis to the upper 500m.

At BATS, there is a long-term positive trend in DIC in all layers (Figure 9b). Reconstructed DIC captures the trend for carbon within the 0-100m and 100m-500m layers, but underestimates DIC in the 1000-2000m layer. DIC also exhibits interannual variability within the top 500m. While both the trend and variability are captured well in the top 100m, between 100m-500m the interannual variability is dampened.

At HOT, DIC in the upper 100m is characterised by interannual variability, which the reconstruction captures. Between 100-500m DIC has a strong positive trend as well as interannual variability. The recontruction captures both the trend and most of the interannual variability at this depth level.

Lastly, for a global view of the reconstruction we provide maps of the Norwegian ESM DIC changes and their reconstruction analogues. Reconstructed interannual carbon inventory changes, shown here to be the top 2000m DIC changes between year 2013 and year 2014, broadly match those in the model (Figure 9c-d). The reconstruction captures the increase in carbon in the equatorial Pacific, the decrease in carbon in the subtropical North Pacific, and much of the smaller-scale structure. Globally, the reconstruction has a spatial pattern correlation of $\rho = 0.48$ for these interannual changes, due partially to the choices for the cutoff radii.

Longer-term carbon changes in the Argo period are marked by widespread increases, particularly in the Northern Hemisphere, and a DIC decrease in the Pacific (Figure 9e-f). The reconstruction captures the near-global increase in DIC and the decreases in DIC in the north and equatorial Pacific. The finer structures within the decadal DIC, such as the regions with the strongest DIC increases, are not fully captured by the reconstruction. The pattern correlation of $\rho = 0.71$ reflects the heightened role of atmospheric $pCO_2$ changes in setting carbon inventories on decadal timescales.

## 6   Discussion and conclusions

In this study a new method for reconstructing upper-ocean carbon using observations of ocean temperatures and salinities is presented. While the ocean plays a large role in determining the partitioning of carbon in the Earth system, sparse observations inhibit a full characterisation of ocean interior carbon. Using synthetic profiles and creating mapped reconstructions of model truths, we have explored how wide-spread synthetic observations of temperature and salinity representative of autonomous sampling programmes can provide global information about ocean carbon. Through reconstructions of CMIP6 model carbon fields, near-surface carbon can be reconstructed using synthetic Argo observations due to consistent controls of solubility, alkalinity, and undersaturation on the carbonate system. The method retains skill for reconstructions down to 1000m. Uncertainties in the model covariance fields below 1000m reduce the skill in which irregular hydrographic measurements can be used to reconstruct carbon changes.

## 6.1 Near-surface carbon reconstructions

When considering the carbon system in the near-surface ocean, the correlation fields between atmospheric $pCO_2$, ocean temperature, ocean salinity, and ocean carbon reflect first-order controls (Fig. 2). Increases in atmospheric $pCO_2$ are correlated with increases in ocean heat and carbon due to the chemical and thermal disequilibria created by emissions. Increases in ocean salinity are broadly correlated with increases in ocean carbon due to the impact of alkalinity on solubility. Regional variations in the correlations between temperature and ocean carbon can be decomposed as a sum of a response related to added carbon from emissions and a residual; the residual correlation reflects the impact of temperature on $CO_2$ solubility in seawater. Within all these correlation fields there are regional variations. The structure in the correlations between temperature and DIC and salinity and DIC attain their strongest values in different regions, and the cross-correlations between temperature and DIC are generally more moderate, indicating that the information provided by these observations are complementary.

The strength and structure in the covariance fields results in optimal weights for $pCO_2$, temperature, and salinity that reflect global first-order controls of solubility and alkalinity (Fig. 4). For this linear model, using co-located temperature and salinity observations along with globally averaged atmospheric $pCO_2$ concentrations reduces the RMSE of the reconstructed DIC content by over 75% on average, relative to a reconstruction based purely on a time-invariant climatology (Fig. 5). In addition to replicating the long-term trend, the reconstructions are able to reproduce local interannual and decadal variability, capturing over 60% of the detrended upper-ocean carbon signal. When considering the detrended carbon inventory, the sensitivity tests show an average relative RMSE reduction of 50% in most regions. The lower skill in reconstructing interannual and decadal variability is expected as the carbon system is nonlinear and controlled by other aspects such as circulation changes and ocean biology, which are not explicitly included in the variables used to calculate the covariance fields.

We have also explored whether the use of irregular temperature and salinity observations impacts the capacity to reconstruct ocean carbon. In theory, observations do not need to be co-located with the desired reconstruction, as the Ensemble Optimal Interpolation scheme can include spatial covariances; however, there is the possibility of extra errors in the covariance fields from poorly represented spatial variability. In our sensitivity tests, we find that using irregular observations consistent with current Argo coverage can be combined with the CMIP6 covariance fields and provide similar levels of skill to the reconstruction. When taking the average RMSE reduction across our sensitivity tests, there is a reduction of over 60% in most regions when considering the full carbon signal and a reduction of over 40% in most regions when considering the detrended carbon signal (Fig. 6b,d). However, regions such as the Southern Ocean and the North Atlantic show substantial error increases relative to the climatological first guess in some of the sensitivity experiments (Fig. 6a,d). These increases in errors are found particularly when considering the detrended signal (i.e., the signal from interannual and decadal variability). The regions that show the largest error increases are those with high mesoscale ocean eddy activity, and are mainly in the sensitivity test that reconstructs ocean carbon within MPI-ESM1.2-LR. MPI-ESM1.2-LR has a lower nominal resolution than the other models in our ensemble, which suggests that the models' representation of mesoscale-scale ocean dynamics and their correlation lengthscales are important factors for any real-world carbon reconstruction.

## 6.2   Reconstructions of interior carbon

The Argo programme measures temperature and salinity to 2000m, which potentially allows for the majority of ocean carbon changes to be captured by our reconstruction. However, within the interior, uncertainties in ocean circulation, ventilation pathways, and regenerated carbon pools can limit the skill of our reconstruction. We find that the optimal solutions for our interior carbon reconstructions are highly depth-dependent. Below 1000m, the use of only temperature and salinity as oceanographic variables may be insufficient as the optimal coefficients become highly correlated.

Sensitivity experiments suggest that the heightened uncertainties in the CMIP6 models reduce the available skill in reconstructing carbon from only temperature and salinity observations at depths greater than 1000m. This increase in errors below 1000m arises from the ensemble construction, particularly the inclusion of the UKESM1. When the UKESM1 is excluded, the remaining models are more easily able to replicate each other's carbon fields. Before creating a real-world carbon reconstruction, further investigation is necessary to determine whether the uncertainties provided by outlier models such as

MPI-ESM1.2-LR and UKESM1 are physically-based or an artefact of model architecture or spin-up procedure.

When considering global reconstructions within the upper 2000m, we find that the top 500m is important for capturing both the trend and variability in ocean carbon. Within the top 500m the method is able to reproduce the trend and variability with skill, and synthetic tests with the NorESM indicate that patterns of interannual and decadal DIC changes can be captured.

## 6.3   Flexibility of Ensemble Optimal Interpolation

While we have opted to use only temperature and salinity as our ocean observations for this scheme, the ensemble optimal interpolation method is flexible and can take additional oceanographic variables such as oxygen, pH, nutrients, or chlorophyll. With increased coverage of these variables from campaigns such as the Southern Ocean Carbon and Climate Observations and Modeling programme (SOCCOM, Johnson et al. 2017) and Bio-Argo (Claustre et al., 2020), there is the potential to include these biologically-affected variables. For the high-latitude oceans where there are few Argo profiles, sea ice observations could

lend additional information on upper-ocean carbon through the impact of sea ice on air-sea gas exchange.

With each new variable it is possible to quantify the amount of added information and test the ensemble optimal interpolation method at various stages by examining the covariance fields and uncertainties, and conducting sensitivity tests in a similar way as has been done in this work. As more complex reconstruction schemes such as the machine learning algorithms of Landschützer et al. (2016) use these biogeochemical observations as well, including them in the Ensemble Optimal Interpolation

scheme could enable a direct comparison between linear and non-linear mapping methods and can help to quantify the merits of linear versus non-linear assumptions when reconstructing ocean interior carbon.

## 6.4   Usage of CMIP6 covariance fields

The Ensemble Optimal Interpolation scheme relies on some important assumptions. Firstly, the covariance fields assume that the processes relating ocean carbon to other variables are stationary. We have used historical CMIP6 experiments to calculate

the covariances fields, so these covariances should be able to represent ocean carbon behaviour under current-day carbon

emission forcing; however, under low or negative emissions the relationships between atmospheric $pCO_2$, ocean temperature, and ocean carbon changes will likely change due to hysteresis from continued heat uptake. Additionally, we have chosen to focus on annual average and depth-integrated carbon content in order to focus on the physical carbon response. It is also possible to consider seasonal or monthly variability by calculating covariance fields for each month or season, as is done for
surface $pCO_2$ and heat content (Smith and Murphy, 2007; Jones et al., 2015). In the current format, months or seasons that are poorly observed will have solutions that return towards climatological inventories. However, the system can be set up in such a way that lagged covariance fields propagate temperature and salinity information from observed to unobserved months as well as locations. The lack of wintertime carbon observations has created significant biases in machine learning products, as enhanced winter outgassing in polar waters is not present in the training data (Bushinsky et al., 2019). As the response to
gaps in data are sensitive to the reconstruction method, the Ensemble Optimal Interpolation reconstructions with seasonal data could provide added insight as to how information on the carbon system is best propagated to unobserved regions and seasons.

   The use of a model ensemble rather than a single model allows for quantification of some of the uncertainties within the covariance fields. Our model ensemble has been constructed to consider both inter-model and intra-model uncertainty. We have been able to explore the sensitivity of this method to the ensemble makeup through out-of-sample reconstructions, and
assessing the minimum, mean and maximum RMSE improvements. These statistics are model-specific; for outlier models such as the MPI-ESM1.2-LR for the upper 100m and the UKESM1 for the 1000m-2000m layer, the poor reconstructions reduce the average skill. Including outlier models should dampen the solution towards a climatology in regions where there are physical uncertainties. The ensemble makeup itself is limited only by the amount of CMIP6 model runs available, and so for future work we will include additional models with fewer realisations.

The improvements in the reconstructions indicate both that the models have regions with well-defined and strong correlations that lead to high improvements as well as regions where the models are relatively uncertain in their relationships (Figs. 5, 6). While these sensitivity experiments and synthetic reconstructions allow for useful insight into the capacity of our reconstruction method, some level of uncertainty remains from the use of coarse-resolution CMIP6 models. Ocean models underestimate decadal variability in anthropogenic carbon (DeVries et al., 2019) and CMIP6 models exhibit biases in the Revelle factor
(Terhaar et al., 2022). More iterative Ensemble Optimal Interpolation techniques can be used to reduce the errors from the model ensemble by recalculating the covariance fields after observations have been included in the analysis (Smith and Murphy, 2007; Smith et al., 2015; Allison et al., 2019).

## 6.5   Moving from synthetic to real-world reconstructions of ocean DIC

Our tests using synthetic reconstructions have suggested that a large amount of ocean carbon variability can be reconstructed
using only hydrographic observations. As such, this work provides a theoretical basis for applying linear reconstruction methods to real-world Argo profile data to create an independent estimate of the recent ocean carbon inventory. The transition from synthetic reconstructions to real-world reconstructions involves careful understanding of observational errors.

   Unlike model output, which can be considered to be "perfect" outside of rounding and regridding errors, observational data have multiple sources of uncertainty, such as sensor errors, the representativeness of area- and time-averaged fields from

Lagrangian observations, and errors in the background climatology fields. While there is rigorous quality control for Argo observations, the representation errors and climatology errors may contribute a substantial amount of uncertainty to the real-world reconstruction. Reconstructions of monthly salinity and temperature have used a parameterised observation error covariance fields (Smith and Murphy, 2007). Future work towards a real-world reconstruction will include a biogeochemical extension of this parameterisation.

In conclusion, we have presented a method that draws upon first-principle relationships between temperature, salinity, and carbon to reconstruct ocean carbon changes from Argo style observations. We find that, for what can be considered the mixed layer, the CMIP6 models show consistent first principle constraints of solubility, alkalinity, and undersaturation, leading to accurate reproductions of trends and variability in the model world. When considering current-day distributions of Argo profiles, reconstructions that use multiple nearby observations are able to still reconstruct the trend and variability in carbon content. Below the mixed layer, the optimal coefficients become more spatially heterogeneous, and out-of-sample tests indicate lower skill in reconstructing carbon changes. However, because carbon uptake occurs at the atmosphere/ocean interface and the response within the top 500m dominantes the upper-ocean response, reconstructions of the upper 2000m using Argo-type observations can recreate patterns of interannual and decadal variability. The results of our work provide a strong theoretical basis for using state-of-the-art autonomous hydrographic observations to supplement sparse interior carbon coverage. This proof-of-concept
work shows that there is strong potential in using these measurements to create a new, independent estimate that can be used alongside reconstruction methods to better understand ocean carbon variability and its controls.

*Code availability.* The code to de-drift, regrid, concatenate, and analyse CMIP6 model output for synthetic DIC reconstructions is available in a Zenodo reposity (DOI: 10.5281/zenodo.7782759).

## Appendix A: Breakdown of covariance fields into $pCO_2$ and non-$pCO_2$ terms

It is possible to decompose the changes in ocean temperature, salinity, and DIC from their time-averaged states ($T', S', DIC'$ from Section 2) into terms proportional to atmospheric $pCO_2$ and a residual:

$$T'(x,y,t) = \alpha(x,y) \, pCO_2'(t) + T_a(x,y,t) \tag{A1}$$

$$S'(x,y,t) = \beta(x,y) \, pCO_2'(t) + S_a(x,y,t) \tag{A2}$$

$$DIC'(x,y,t) = \gamma(x,y) \, pCO_2'(t) + DIC_a(x,y,t), \tag{A3}$$

where $\alpha, \beta, \gamma$ are the least-squares coefficients against changes in globally uniform atmospheric $pCO_2$ changes and $T_a, S_a, DIC_a$ indicate the residuals for these decompositions.

While the decomposition into $pCO_2$ and non-$pCO_2$ terms is not inherently physically based, this breakdown may help us understand how the complex structure in the correlation fields in Fig. 2 arise. Using covariance identities, the covariance between, for instance, $T'$ and $S'$, is thus the sum of the covariances between the addends:

$$cov(T', S') = cov(\alpha \, pCO_2(t), \beta \, pCO_2(t)) + cov(\alpha \, pCO_2(t), S'') + cov(T'', \beta \, pCO_2(t)) + cov(T'', S''). \tag{A4}$$

The first term is a function of the variance of $pCO_2$, scaled by the product $\alpha\beta$. If the term proportional to atmospheric $pCO_2$ were orthogonal to the residual term for each of these decompositions, the covariances $cov(pCO_2, X_a) = 0$ for any variable $X$. An ordinary least squares solution does not lead to orthogonal components, but the covariances between atmospheric $pCO_2$ and $DIC_a, T_a$, or $S_a$ are small (shown as correlations in Fig. A1, panels d h and i) and can therefore be ignored to first order.

Thus for these variables we ignore these terms as second-order components and can thus approximate the covariance as:

$$cov(T', S') \approx \alpha\beta \, var(pCO_2) + cov(T_a, S_a). \tag{A5}$$

With this decomposition of the covariance fields into $pCO_2$ and non-$pCO_2$ terms, we have the added benefit of approximating the covariance fields between $pCO_2$ and one of $T', S'$, or $DIC'$ through a global scaling of the coefficient terms $\alpha, \beta$, or $\gamma$, respectively.

Scaling the decomposition by the product of the standard deviations allows us to explore how the correlation fields arise in Fig. 2. We compare the original correlation fields from Fig. 2 with their decompositions according to (A4) in Fig. A1. The approximation errors show how well the approximation in (A5) reflects the full correlation field.

For the correlations between $T'$ and $S'$ and $S'$ and $DIC'$, the signal is dominated by the term associated with $cov(T_a, S_a)$ and $cov(S_a, DIC_a)$, respectively. We therefore understand these covariance and correlation fields to be set by the variability

not forced by carbon emissions. Conversely, the correlation field between $T'$ and $DIC'$ appears to arise almost equally from the $pCO_2$ scaling and the $cov(T_a, DIC_a)$ terms. Thus we provide both terms in Section 3 to gain insight as to how the covariance fields reflect both the combined response to added carbon in the Earth system as well as the response to climate variability.

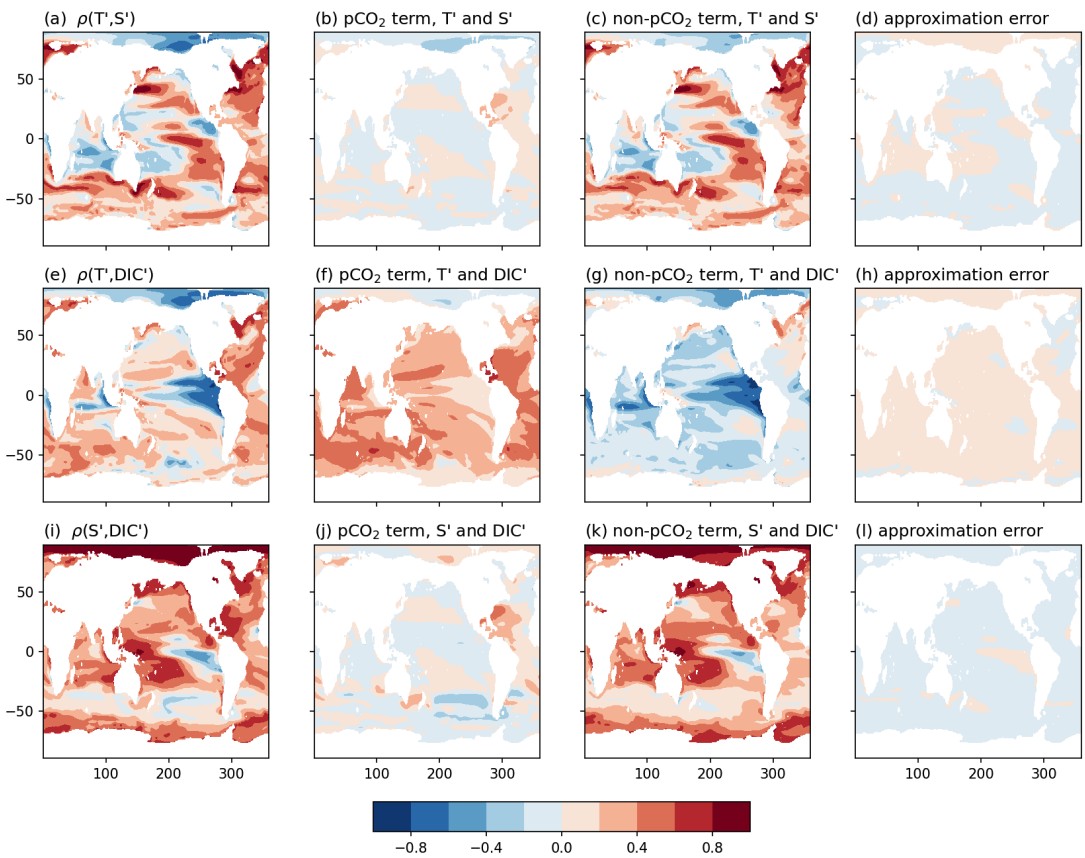

**Figure A1.** Correlation fields between non-pCO₂ variables $T', S'$, and $DIC'$, with their terms from (A5) and approximation errors to measure the level of mismatch between (A4) and (A5).

*Author contributions.* All authors contributed towards the conceptualisation of the work. KET created the method and analysis with input from RGW, AK, and DMS. KT wrote the manuscript draft, and all authors reviewed and edited the manuscript. All authors signed off on the manuscript for submission.

*Competing interests.* The authors declare that they have no competing interests.

*Acknowledgements.* KET received support from the Leverhulme Trust via the Leverhulme Research Centre for Functional Materials Design. Authors KET, RGW, and AK received research support from the U.K. Natural Environmental Research Council (Grant number NE/T007788/1). DMS was supported by the Met Office Hadley Centre Climate Programme funded by BEIS and Defra. The authors thank 2 anonymous referees for their constructive feedback.

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
