# Peer review of "Reconstructing ocean carbon storage with CMIP6 models and synthetic Argo observations"

_Biogeosciences, 2022_

## Author Response (AR1)

**Response to reviewers on "Reconstructing ocean carbon storage with CMIP6 models and synthetic Argo observations"**

To the editor and reviewers:

On behalf of myself and my co-authors, I would like to thank you for sending our manuscript "Reconstructing ocean carbon storage with CMIP6 models and synthetic Argo observations" out to review. We have found the reviewers' comments insightful and thought-provoking, and we would like to thank both reviewers for taking the time and effort to carefully consider the manuscript. We hope that we can iterate upon this manuscript, taking all the comments into account, and produce an improved version for *Biogeosciences.*

Before answering the reviewers' comments and questions line by line, we would like to provide an overview to our response. Both reviewers raise important points on the scope of the study, particularly regarding two points: 1) the decisions made about depth level chosen for our reconstruction, and 2) the lack of a real-world construction using existing Argo profiles. We have extended the manuscript to include an analysis over the upper 2000m and a longer discussion on the transition from synthetic to real-world reconstructions to respond to the reviewers' comments.

**1. Extension of synthetic reconstructions with depth**

Both reviewers have questioned the restriction of our reconstruction method to the top 100m carbon response. Upon reflection, we agree with the reviewers that this choice is too restrictive. The Ensemble Optimal Interpolation method, being an off-line and linear method, is highly flexible, and extensions with depth are straightforward in our synthetic reconstructions. We originally chose the depth horizon of 100m as a test case, but have extended the method to reconstruct carbon within the upper 2000 m. This choice of the upper 2000 m is to maximise the data coverage of the ocean and to be consistent with the maximum depth range for Argo profiles.

It is possible to reconstruct carbon up to 2000m in two ways, by either using temperature and salinity anomalies integrated over the entire depth range, or by breaking up the water column into layers. We have opted for the second method, as different depths have different controlling processes and will therefore have different imprints on the covariance fields between carbon, temperature, and salinity. We have extended the method by considering reconstructions for three additional layers to the 0-100m layer presented in the first manuscript: 100m-500m, 500m-1000m, and 1000m-2000m. For simplification, we present here the optimal coefficients for atmospheric $pCO_2$ and co-located temperature and salinity, alongside average improvements calculated from our sensitivity tests.

[Figure]

*Figure 1 Optimal weights for interior ocean DIC as a combination of atmospheric pCO2 (top row, panels a-c), ocean temperature (middle row, panels d-f), and ocean salinity (bottom row, panels g-i). The weights were calculated to solve for DIC in various interior layers: 100-500m (left column, panels a,d,g), 500m-1000m (centre column, panels b,e,h), and 1000-2000m (right column, panels c,f,i).*

The optimal coefficients for solving for interior DIC using atmospheric $pCO_2$ and co-located temperature and salinity anomalies show depth-dependency in their structure (Figure 1). Coefficients for atmospheric $pCO_2$ (subplots a-c) remain positive over the top 1000m; for the 100m-500m layer, they exhibit local maxima in the subtropical gyres, whereas for the 500m-1000m layer the maxima are located in the regions of mode water formation. For the deepest layer of 1000m-2000m (subplot c), the coefficients are near-zero for most of the ocean and negative in the regions with oldest waters (i.e., the equatorial Indian and equatorial and north Pacific). The strongest impact is found in the North Atlantic, consistent with the ventilation of North Atlantic Deep Water and transport equatorward in this layer.

As for the top 100m, temperature coefficients (subplots d-f) are generally negative and salinity coefficients (subplots g-i) are generally positive for the ocean interior. The regional structures of both of these terms differs from those for the upper 100m, most noticeably in the 500m-1000m salinity coefficients (subplots e,h). We also note that for the deepest layer of 1000m-2000m, both temperature and salinity coefficients exhibit dipole behaviour in the Pacific Ocean and are near-zero elsewhere (subplots f,i). The dipoles for salinity and temperature coefficients are similar, suggesting that the information provided by temperature and salinity profiles in the deep Pacific is no longer independent.

[Figure]

*Figure 2 Ensemble average relative RMSE reduction ε, as measured across the out-of- sample tests as described in section 4, for (a) integrated DIC between 0-100m, (b) integrated DIC between 100-500m, (c) integrated DIC between 500-1000m, and (d), integrated DIC between 1000-2000m. Subfigures (e-h) are as above, but consider only the detrended aspect of integrated DIC in the same layers.*

For the reconstructions of individual layers, we find that the overall skill in the reconstructions decreases as we move further into the ocean interior (Figure 2). Up to a depth of 500m, the covariance fields between ocean DIC, temperature, salinity, and atmospheric $pCO_2$ is good almost everywhere; the reconstruction improves upon the climatology (subplots a-b), while there is improvement in the representation of variability in most regions (subplots e-f). Below 1000m, we find large areas where the reconstruction performs worse than the climatological first guess, particularly in the Pacific (subplots c-d and g-h). The regions that show skill are those in which mode water formations maintains similar relationships between DIC and hydrography via solubility and alkalinity arguments.

However, as Reviewer #1 has noted, the deeper ocean contributes a small fraction of the overall change in ocean carbon. When conducting the same sensitivity tests on reconstructions of the upper 2000m (Figure 3, not included in manuscript), the reconstruction errors in the 500m-1000m layer and 1000m-2000m layer reduce the overall skill, but the method still provides more information over the climatology outside of a few points.

We have extended the manuscript considerably to include results with depth. In addition to the work calculating the optimal coefficients and conducting similar out-of-sample tests with reconstructions at depth, we have extended the NorESM reconstruction to consider changes in the upper 2000m over a modelled "Argo-observed" period (Figure 9 in the manuscript). The extension shows that linear reconstructions can provide skill to interior carbon layers, that there are higher uncertainties and

potential errors associated with these interior reconstructions, and that the skill for column inventories remains substantial due to the vertical structure of carbon inventory changes.

[Figure]

*Figure 3 Ensemble average relative RMSE reduction $\varepsilon$, as measured across the out-of-sample tests as described in section 4, for column integrate DIC down to a level of 0-100m (a,b), 0-500m (c,d), 0-1000m (e,f), and 0-2000m (g,h). Left column shows $\varepsilon$ for the full DIC signal, whereas the right column shows $\varepsilon$ for the DIC signal once a linear trend has been removed.*

**2. Inclusion of real-world Argo floats into a full carbon reconstruction**

Both reviewers also mentioned a degree of disappointment that this method has not yet been applied to Argo. A real-world reconstruction is the overall goal of this work; however, we feel that a full reconstruction should be approached in a second work, as there is an additional substantial amount of analysis to understand how errors in the observations impact the reconstruction. Without conducting this additional analysis, it would be easy to provide a misleading view as to the benefits of applying the method to real Argo data.

Synthetic reconstructions draw upon the assumption that model data is a near-perfect representation of the climate system at any given location and time. Model output should have no errors other than those created from regridding and rounding. However, when considering real-world data, there are additional errors within the observations. These observational errors arise from a combination of:

1. Sensor errors
2. Representation errors, consisting of
    a. Spatial binning errors, such as aliasing of structures such as eddies through repeat sampling with Lagrangian profilers
    b. Temporal binning errors, particularly regarding the estimation of an annual mean state from different months of observations
3. The estimate of the climatological first-guess field

We believe that the errors arising from points 2 and 3 may be substantial. While it would be possible to calculate an estimate by assuming the real-world observational errors are 0, we believe this inclusion could mislead the readers as to the actual impact of the method and have therefore refrained from doing so.

Ultimately, creating a real-world reconstruction includes an *observational error covariance* field that considers errors from the above sources. Reconstructions of salinity and temperature have used a parameterised observation error covariance field; for instance, Smith and Murphy (2007) parameterise the observational error in their temperature and salinity reanalyses to be

$$E^{obs} = {E_{inst}}/{N} + f\sigma^2({1}/{N} + 1) \ ,$$

where $E_{inst}$ is the instrumental error, $N$ is the number of profiles in a grid cell in a given month, and $f\sigma^2$ is the fraction of monthly variability within a grid cell. The authors objectively choose values for $f\sigma^2$ based off extra validation studies.

Based off our knowledge of similar reconstructions and the differences between previous works and our multivariate system for ocean carbon, we would prefer to create our own validation studies against existing carbon observations to quantify these error covariance fields. These validation studies could be used to further compare linear and nonlinear reconstruction methods, as increased wintertime observations have lowered estimates of the Southern Ocean carbon sink (Bushinsky et al., 2019). We propose that we extend the discussion in the manuscript to include a more in-depth description of how the method will be extended to account for observational errors and the complexities involved.

**Responses to individual reviewer comments – Reviewer #1**

(Reviewer's comments are in black, authors' response is in *blue italics*)

The manuscript "Reconstructing ocean carbon storage with CMIP6 models and synthetic Argo observations" by Turner et al. is well written and easy to follow. The figures are well chosen, good to read, and facilitate the understanding of the study. The methods are sound and well presented, although some precisions at a few points would be needed, I think.

However, I am somewhat concerned about the novelty and significance of this work for this journal, although this might well be due to a misunderstanding of the methods. To be sure that no misunderstandings have occurred, I am first summarizing the paper very(!) briefly in my own words. If the authors detect significant flaws, please consider them when reading my following comments.

As far as I understand, the here presented method builds on the assumption that the DIC (averaged over the first 100 m in the ocean) at a point x,y can be expressed as the average DIC from 1955 to 2014 at this point x,y plus a delta DIC. This delta DIC can, for any moment in time, be approximated by a linear combination of delta pCO2, delta T, and delta S at locations more or less close to point delta x,y.

*We thank Reviewer #1 for their thoughtful and constructive review. Our aim with this manuscript is to present an independent method for reconstructing ocean carbon changes. The method is flexible and can be constructed with a variety of choices regarding depth horizon, ensemble makeup, and input parameters – all whilst being able to refer to model covariance fields to understand the scope and impact of any choices within the method. We believe that this flexibility is a strong feature of our approach and is a novel method for reconstructing interior carbon content. Our study provides a framework for a new carbon reconstruction alongside current nonlinear estimates of carbon fluxes (e.g., Landschützer et al., 2014) and seasonal interior DIC (e.g., Keppler et al., 2020).*

My comments are:

- **Long-term trend**

To me, it seems somewhat trivial that the long-term trajectory of DIC in the first 100 m, a depth that is more or less representative of the average mixed layer depth, is mainly determined by the atmospheric pCO2. The provided BATS example is an excellent example (blue dashed line in Figure 7b). By knowing also temperature, which affects solubility, and salinity, which is closely related to alkalinity, at only one point close to the point of interest, slight deviations of this long-term trend can be represented, as well as inter-annual and decadal variability that is not due to changes in primary production or remineralization can be represented. This is nicely represented by the HOT example (blue dashed line in Figure 7c). As the authors describe in their paper, the underlying mechanisms are well-understood: air-sea CO2 flux due to increasing atmospheric pCO2, decreased solubility with increasing temperatures, and more DIC with increasing alkalinity. Therefore, I think that the predictability of DIC in the mixed layer from pCO2, T, and S as well as a background DIC, is not surprising.

*On timescales longer than a few decades, changes in DIC are dominated by the addition of carbon through emissions. The top 100m response is indeed generally a mixed-layer response. T and S observations do add further information for the long-term response, particularly in terms of how they are related to atmospheric pCO$_2$ changes (manuscript figures 2b, 2c, 3a). As the referee*

*suggests, the dominant response is well understood and involves increases in atmospheric $pCO_2$, changes in buffered chemistry, decreasing solubility with increasing temperature, and increasing solubility with increasing alkalinity. The consistency to which these mechanisms are replicated within the model ensemble provides evidence of skill for the reconstructions for near-surface DIC using the model ensemble.*

*However, reconstructing the trend for deeper layers is less straightforward. On extension of the method to include the ocean interior down to 2000m, we find that the optimal solution from the CMIP6 model ensemble reflects the control of $pCO_2$ in regions with well-ventilated waters (Figure 1, subplots a-c). Up to a depth of 1000m, the optimal solutions for temperature and salinity remain consistent with previous solubility and alkalinity arguments, although the structure of the optimal solution is depth-dependent (Figure 1, d-e and g-h). Below 1000m, the optimal solutions for temperature and salinity have higher regional structure and become positively correlated, particularly in the Pacific.*

*To illustrate the role of various depth levels in setting long-term versus interannual DIC signals, we have expanded the original manuscript's Figure 7 (now Figure 9) to show model truth and reconstructed DIC at our depth levels down to 2000m. Globally, the longer-term trend is dominated mainly by the 100m-500m layer, while the 0-100m layer shows the strongest interannual variability (new Figure 9a). On a regional scale these relationships are altered; for instance, at BATS there is a strong linear signal below 1000m, and the 100-500m exhibits strong interannual variability (new Figure 9b). The HOT reconstruction also shows that the 100-500m reconstruction can be important for capturing the trend and variability of regional carbon inventories (new figure 9c).*

- **Variability (detrended signal)**

When the detrended response is looked at, pCO2, which is responsible for the largest part of the trend, should not play any role anymore as annual globally averaged pCO2 has little variability. At this point, the RMSE improvement seems to drop (Figures 5d-f and 6d-f). It remains mainly high in the relatively slow and calm subtropical gyres and becomes smaller in the more dynamical regions like the tropical Pacific, eastern upwelling regions, or the Southern Ocean. In these regions, variability in close-to-surface DIC is likely influenced by a variability in the circulation and hence the upwelling of nutrients and carbon. The negative correlation between DIC and T in the tropical eastern Pacific (2d) due to El Nino is a good example. It would be nice to discuss and analyze a little bit more, why these regions are less predictable. As these regions are usually also the regions with the highest variability in surface ocean DIC, I was wondering how the statement that the RMSE of the detrended signal was reduced by 60% was calculated. Did you calculate the area-averaged eta from Figures 5e and 6e? Or did you compare the detrended true signal of globally averaged DIC to the predicted signal of globally averaged DIC? I am not sure which way is better as both have potential pitfalls. Taking the area-weighted average of eta gives potentially to strong emphasize to regions with little variability, as the subtropical gyres, that contribute only little to the global ocean DIC signal. However, by taking the area-averaged DIC first, errors in different directions may globally compensate. Maybe it would be best to use the local eta at each point x,y and weight them by the true DIC variability at that point x,y?

*Thank you for your comments. For the detrended response, we remove a linear trend from the 60-year model runs taken for each of the out-of-sample tests. As atmospheric $pCO_2$ rises approximately exponentially during this longer period, the detrended response retains some second-order behaviour in $pCO_2$.*

*The regions which show the greatest RMSE improvement (when considering the detrended signal) are regions in which the models have similar ratios between temperature, salinity, and DIC variability, even if they differ on the structure or power of modes of variability such as ENSO. For instance, the negative correlations between DIC and T found in the models results in detrended RMSE improvements that, while less than those for the western basins, is greater than those for the shadow zones in the Pacific and Atlantic. Additionally, the dynamically active North Atlantic shows strong improvements in the detrended RMSE.*

*Regions with minimal RMSE improvements are marked by moderate correlations between temperature and salinity, which suggests that the information provided by temperature and salinity is less orthogonal in those regions. This could be a result of dynamical uncertainty within the model ensemble or an underdetermined system that would benefit from the addition of other ocean tracers. As our reconstruction method is statistical rather than dynamic, understanding the causes of these uncertainties requires a detailed examination of the controlling processes within each model before relating back to the covariance fields.*

*In the submitted manuscript we avoided creating area-weighted averages of eta. The formulation of the relative RMSE reduction eta includes a normalisation by the RMSE from the first-guess climatology field, which is equivalent to the variance of DIC at a given location. We have refrained from creating a global measure for the same reasons the reviewer has pointed out. In the manuscript abstract and show in manuscript Figure 5 we show that in most regions, the reconstruction reduces the RMSE by 60% or more. We have extended the manuscript to include the global RMSE reduction (equivalent to the variance-weighted eta) in Section 3.3: "For reconstructions of global DIC inventories using co-located temperature and salinity observations, the sensitivity experiments show an average RMSE reduction of 93%. When considering detrended DIC inventories, the sensitivity experiments reduce the RMSE by 68% on average."*

*For the ocean interior, the errors in the sensitivity tests coupled with the low variability found in these layers make the quantification of global improvements difficult. Our extended Figure 9 now shows reconstructed carbon for interior layers both on global and regional scales.*

- **Significance of the first 100 m for the long-term ocean carbon sink**

Another point that I am not sure about is the importance of this analysis for the global ocean carbon sink. How much of the additional DIC is in the first 100 m? The Global Carbon Budget 2021 (https://essd.copernicus.org/articles/14/1917/2022/essd-14-1917-2022.html) estimates that from 1960 to 2020, 115 Pg C has entered the global ocean. Recent estimates from observationally constrained ESM output suggests that it might be a bit more. Thus from 1960 to 2014, the ocean has taken up very approximately ~100 Pg C. The upper ocean carbon inventory shoes an uptake of ~14 Pg C (tried to read that number from Figure 7a). Thus, the upper ocean is 'only' responsible for 14% of the global ocean carbon sink. Wouldn't it be more important to quantify how much is transferred from the surface ocean to the interior ocean, i.e., subducted below the mixed layer (https://agupubs.onlinelibrary.wiley.com/doi/full/10.1002/gbc.20092, https://agupubs.onlinelibrary.wiley.com/doi/full/10.1002/2015GL065073). Recent studies indicate that salinity and temperature might indeed be able to reproduce the interior ocean Cant (https://www.science.org/doi/10.1126/sciadv.abd5964, https://www.nature.com/articles/s41586-020-2360-3, https://bg.copernicus.org/articles/18/2221/2021/bg-18-2221-2021.html#&gid=1&pid=1, and https://bg.copernicus.org/preprints/bg-2022-134/). However, the interior ocean transport might make this much harder. The authors mention that this will happen in the next step, but I think it could/should also be incorporated in here if(!) the long-term trend remains a focus of this study.

*Originally, we chose the top 100m as a test case to show how the linear optimal interpolation system performs by drawing upon the covariance fields. Upon reflection of the comments for both reviewers we agree that this choice was too limited. Instead we have extended the method to a depth of 2000m. Much of the carbon uptake and variability is likely to be in the upper 2000m of the ocean, so this extension should better capture the ocean carbon sink response. Our analysis highlights the difficulties that the reviewer brings up, where uncertainties in ocean transport and a less surface-driven response to emissions reduces the reconstruction skill.*

*We have also updated the original manuscript Figure 7 (now Figure 9) to include a full reconstruction with depth using the Norwegian ESM in all panels. For simplicity we use cutoff radii for each depth level, going as far as to us only co-located profiles below 1000m where there are smaller signals and larger errors present in our sensitivity tests (see Figure 2 in this response). The dominance of the upper 500m is noticeable for the trend and variability in ocean carbon content. For regional carbon inventory changes such as those found at BATS, our reconstructions show the importance of even deeper carbon content changes.*

- **Significance of the first 100 m for the variability of the ocean carbon sink**

Figure 7a suggests, however, that the fist 100 m might be very interesting for the inter-annual or decadal variability. In Figure 7a, you show the variability of the upper ocean 100 m DIC. How much of this is caused by the air-sea CO2 flux and how much by transport to the interior ocean. Could the variability in the air-sea CO2 flux maybe largely be derived from variability in the ocean T and S? Or in the variability of subduction on these time-scales? Can your method provide an estimate of the subduction of DIC on an annual basis using the surface ocean air-sea CO2 flux estimates in combination with your estimates? The example of HOT in NorESM is striking as it suggests that the upper-ocean has taken up almost no carbon between 1990 and 2014 despite an increase in atmospheric pCO2.

*From covariance fields alone it is difficult to disentangle the various flux drivers of carbon content; however, with current estimates of pCO$_2$ fluxes from other products (such as the machine learning methods in Landschützer et al., 2014) alongside reconstructions at various depth horizons, it would be possible to create an estimate of global subduction rates for carbon. In a model analogue, Lauderdale et al. (2016) have explored the drivers of atmospheric CO$_2$ uptake using the MITgcm, and include discussion on the role of advection, diffusion, and biological activity on upper ocean carbon content.*

*We note that the Earth system model runs used for this work are not reanalysis datasets but historical simulations with their own timings for climate modes of variability. Thus, the hiatus in upper-ocean carbon at HOT in the NorESM is not necessarily indicative of real-world conditions in the 1990s and 2000s, but it is still an interesting note that these local hiatuses are present in the model data.*

- **Proof-of-concept without application to real data**

After having read through the manuscript, I was really disappointed that this new method is not yet applied to observations right now. There is much potential, and I understand the idea of making two publications, one for 'proof-of-concept' and one for the application, but now it seems as if something is missing in this paper. If no institutional, or PhD-related restrictions exists, I would recommend to also present the application here.

*The implementation of Argo data into the Ensemble Optimal Interpolation method is non-trivial, as observations contain their own errors, both in terms of instrumental errors and (in our view, more importantly) errors in how binned and averaged observations represent the desired input variables (for this setup, annual average temperature and salinity fields). We have outlined the difficulties in applying the method to observations in the first part of our response.*

*Argo sensor measurements have relatively small errors, especially when restricting profiles to those that have passed rigorous quality control. However, Argo observations are snapshots rather than mean values over space and time. We will need to carefully consider:*

- *The impact of observations that cover only some months of the year, and particularly those that might be biased towards specific seasons. This bias may be random but it may also be inherent in the Argo programme setup. For instance, the presence of seasonal ice cover will prevent measurements from being taken in winter months.*
- *The aliasing of small-scale structures not present in coarse CMIP6 models with Lagrangian Argo floats. If an Argo float remains within the core of an eddy, its monthly fields may reflect anomalous conditions not representative of the relatively large area we aim to average across.*
- *Sensitivity tests against existing carbon datasets such as GLODAP or time series sites to quantify relative uncertainties from different aspects. Particularly in the interior, validation against GLODAP and time series sites may be important in setting the optimal localisation method.*

*We have provided a short outline of these considerations in our discussion. We have included our logic for this choice in Section 6.5: "Moving from synthetic to real-world reconstructions of DIC."*

My overall recommendation would therefore be to de-emphasize the long-term trend, focus on the variability, apply the method to observations, compare it to estimates of the air-sea CO2 flux, and try to address the subject of the decadal and inter-annual variability of the ocean carbon sink.

As you already have the models, you might even be able to draw conclusions why and where models do not capture the variability. Are the regions with the largest variability in the air-sea CO2 flux also the regions with the largest variability in the upper ocean DIC?

If the focus is more on the detrended signal, the subsurface ocean would not fit anymore and could be left for another study.

While this sounds somewhat negative, I am in strong favor of publishing this manuscript in Biogeoscience but with an application to observations. If not applied, maybe GMD is more suited for 'proof-of-concept' studies? But that is an editorial decision.

*Thank you for your thoughtful review. We will provide an extension for reconstructing ocean carbon behaviour at depth in order to mitigate concerns about how well the upper ocean represents the global ocean carbon response. With this extension we believe we can discuss both the long-term behaviour and interannual behaviour more completely. We will provide context for how modelled DIC behaves with depth (i.e., which layers contain most of the forced signal vs. interannual variability).*

*As the reconstruction uses a limited number of inputs, regions where variability is not captured are not necessarily regions where the models have a mismatch in the variability. There are areas where*

*the models do not capture variability, but our reconstructions are likely more limited by the lack of other tracers to describe ventilation rates or biological activity. Understanding the drivers of the model biases in ocean carbon is an important and difficult question, and likely requires diagnostics of overturning as well as the drivers of pCO₂ fluxes (such as in Lauderdale et al., 2016).*

**Minor comments:**

Line 22, page 1: While the Global Carbon Budget summarizes the numbers, I prefer to give credit to the original studies (for example: https://bg.copernicus.org/articles/10/2169/2013/, https://agupubs.onlinelibrary.wiley.com/doi/full/10.1002/2013GB004739, and recently https://bg.copernicus.org/preprints/bg-2022-134/)

*Agreed. We will update the introduction to include the original studies. We will retain the GCB figure as well as it contains an estimate of anthropogenic fossil fuel emissions and land use changes.*

Line 25, page 2: Gruber et al. is about compound extreme events and rather summarizes literature. I would recommend to replace it by this earlier study on global ocean acidity extremes (https://bg.copernicus.org/articles/17/4633/2020/) and this regional study (https://www.nature.com/articles/s43247-021-00254-z) that provides one example in detail.

*Thank you for this correction, we have made the adjustment in the manuscript.*

Line 57: I think this is the first time that you use the term 'upper ocean'. Maybe define it here.

*We will add a definition for the upper ocean here and define the upper ocean down to a depth level of 2000m due to the depth profiles of Argo data. We have edited the paragraph to include the following (lines 71-73):*

> *In this proof-of-concept study we aim to show the potential skill available in using model covariance fields and Argo-style synthetic measurements to reconstruct carbon content between 0-2000m.*

Lines 60-66 and lines 99-100: Would suggest to add the Southern Ocean studies (https://www.science.org/doi/10.1126/sciadv.abd5964, https://www.nature.com/articles/s41467-022-27979-5), and one study from the Arctic Ocean (https://www.nature.com/articles/s41586-020-2360-3) that show how surface observations (and variations) of S and T can alter the CO2 uptake. Or maybe keep that for later in the Discussion when you describe the potential to go below the surface.

*Thank you for the recommendations. In the introduction we have included the sentence in Line 62:*

> *Anthropogenic carbon uptake in the high latitudes is constrained by salinity and stratification, which can be taken to be proxies for water mass formation (Terhaar et al., 2020; Terhaar et al., 2021; Bourgeois et al., 2022).*

*Emergent constraints provide an interesting statistical perspective on the realism of climate model outputs, and we think that this nicely complements the inter-variable and spatial correlations on which the Ensemble Optimal Interpolation method is built. Particularly in the high latitudes, where the ensemble members show a low level of agreement in our reconstructions, the use of emergent constraints could increase the reconstruction skill.*

Line 112: Does the method account for inter-dependencies between the predictors? Sorry about the question, but I am not very familiar with this kind of method.

*The Ensemble Optimal Interpolation method uses all covariances between input parameters and the desired output parameters to solve for the optimal weights, and so highly-correlated input parameters will have an impact on the optimal weights. We will make this point clearer within the methods section.*

*We have added the line (115-117):*

> *These optimal weights thus describe how information is propagated from atmospheric pCO$_2$ and hydrographic observations to the ocean carbon system, taking into account the interdependencies between input variables.*

Line 132-133: Just wanted to say that I really like that you provided the regridding method here in such detail. Often it is not possible to reproduce data because the regridding software is not named.

*Thank you.*

Line 141: Something is wrong in the sentence, I think.

*We will make the setup of the first idealised experiment clearer by changing the text in the first two sentences to (Line 148-151):*

> *"The first reconstruction method assumes full global coverage of temperature and salinity observations. For these synthetic reconstructions, the ocean inputs to reconstruct ocean DIC are co-located model temperature and salinity anomalies, as well as globally-uniform atmospheric pCO$_2$ anomalies. The resulting system thus has 3 input parameters to solve for ocean carbon at each grid cell."*

Lines 147-154: Just to be sure that I understand it right. When you use the ARGO 2002 coverage, you assume that you have data at each of these points for all years from 1955-2014? Is that right? If yes, how do changes in the temporal coverage effect time series as shown in Figure 7? If you have little data for the first 30 years and then more, you should get a better representation of the variability over time, right? Figure 7c seems to suggest that there is only 1 observation for the blue line, but that means one observation per year in the vicinity of the HOT station, right?

*In the original manuscript, for coverage consistent with Argo profiles in year 2002, we took the observational coverage to be constant in time. This decision was made to better understand how different levels of coverage could capture various levels of change (trends vs. interannual vs. multidecadal variability) and to separate the role of coverage vs covariance errors in determining the reconstruction of skill.*

*Based off the response from both reviewers we have altered the original Figure 7 a-c (now Figure 9) with sample reconstructions from 2002-2015 that use the temporally-varying coverage. In our new figures we only reconstruct 14 years of carbon changes. In Section 2.1, point 2, we have included:*

> *Using this distribution of available profiles, we create reconstructions using time-varying profile locations and test how profile density impacts the reconstruction by setting the distribution as constant in time.*

Section 3: I found this section a little bit difficult to follow. What could help (maybe it doesn't), is to look only at the long-term signal and only at the detrended for the correlations. Than the long-term should give you a correlation between pCO2, T, and DIC. T and DIC because the ocean is warming and taking up carbon (but one does not cause the other) and the other between pCO2 in the atmosphere that causes DIC in the ocean to increase. For the detrended signal, I would expect no correlation between pCO2 and DIC as the pCO2 trend is gone. T and DIC should be negatively correlated (like shown in Figure 3b). And salinity should be positively correlated. A little bit along the lines of Figure 2 in https://www.nature.com/articles/s41467-022-32120-7

*During our creation of the Ensemble Optimal Interpolation method, we experimented with creating a reconstruction of the DIC signal with a linear trend removed. We found that temperature and DIC remained strongly positively correlated for most of the ocean due to the second-order increase of carbon emissions. We find the system reflects the correlations mentioned by the reviewer when removing either a term proportional to atmospheric pCO₂ or when including atmospheric pCO₂ as an input. We have opted for the latter option as we believe this setup remains consistent with the climatology first guess fields found in ocean temperature and salinity EnOI methods. We also believe that this decision increases the transparency of the method, particularly considering that models have biases in their Revelle buffer factors.*

*However, we recognise that this decision leads to a complicated system in terms of the number of covariances involved in setting the optimal weights, and the dissection required to understand the covariances between temperature and DIC. We have reframed Section 3 into smaller blocks to enhance readability, with headers focusing on correlations involving pCO2, the correlation between DIC and temperature (which uses a decomposition to understand the role of the solubility effect versus the combined addition of heat and carbon into the Earth system), and the remaining correlations with salinity.*

General: Did you think about including the area of sea ice in each cell as well? As this blocks the air-sea gas exchange to some extent, it might be a powerful predictor in the high-latitude oceans. Just a guess.

*We did not include sea ice area for the high-latitude reconstruction because the extent of Argo floats is limited to ice-free regions. The presence of sea-ice could provide useful information about air-sea carbon fluxes, particularly when the predicted information could be examined alongside more recent under-ice profiles of temperature, salinity, and potentially also DIC.*

*In our discussion on how the method could possibly be extended to other observations such as oxygen, pH, and nutrients, we will include a statement on how satellite observations of sea ice could be added to provide information on the communication between the ocean and the atmosphere. In Section 6 we have included (Line 492-493):*

> *For the high-latitude oceans where there are few Argo profiles, products of sea ice coverage could lend additional information on upper-ocean carbon through the impact of sea ice on air-sea gas exchange.*

Section 6: While I completely understand the reason to only choose Nor-ESM to make these tests, I would be curious how other models perform.

*Section 4 implicitly includes an evaluation of how the method performs on other models. In Section 4 we have conducted sensitivity tests, in which models within the ensemble are omitted, the covariance*

*fields are re-constructed, and then the out-of-sample models are reconstructed. The figures in Section 4 have been chosen to illustrate the range of responses within the ensemble and to illustrate how models add confidence or uncertainty to the method.*

*Once a model is included in the ensemble, the method can reproduce that model's DIC with greater accuracy. We find this accuracy to be misleading when considering real-world applications in which all the models will have shortcomings and biases. Thus in Section 6 we use the full ensemble as described in section 2 to reconstruct a completely independent model. In our further work including real-world observations, we will likely include the NorESM (as well as other ESMs) in our covariance calculations so that a more complete range of model uncertainty is considered within the covariance fields.*

*To aid the reader in connecting these two sections, in the manuscript we will include a statement in the beginning of Section 6 to describe out the reconstruction of the NorESM is an additional out-of-sample reconstruction in line with the analysis in section 4. We have included the text (line 391-393):*

> *This reconstruction is an additional out-of-sample reconstruction, similar to those made for the error reduction statistics in Sections 4 and 5 but uses covariance fields constructed from the entire CMIP6 model ensemble.*

**Responses to individual reviewer comments – Reviewer #2**

(Reviewer's comments are in black, authors' response is in *blue italics*)

In this study Turner et al. reconstruct the ocean carbon storage from first-order relationships between temperature, salinity and atmospheric CO2 and Dissolved Inorganic Carbon (DIC). The authors use a set of CMIP6 models to assess these relationships and estimate the covariance fields. The inferred statistical relations are then used to reconstruct the carbon storage from hydrographic pseudo-observations. In order to test the capabilities of this approach, two sampling methods are proposed: 1) a complete coverage using CMIP6 co-located observations and 2) Irregular sampling consistent with Argo profiles. While co-located observation (1) is taken as a sensitivity test, the irregular Argo-style observations (2) show the potential of this method to use real Argo measurements to reconstruct the carbon storage. Both of the sampling methods offer a significant improvement compared to the reconstruction based solely on the climatological mean. The study is well-written, presented in a structured logical manner and the results and their implications are easy to follow.

This study represents a significant advancement within the field, offering a powerful method to understand both the spatial and temporal variability of DIC. This method could not only be used to reconstruct the carbon storage from real hydrographic measurements but also, the resulting DIC fields could be used to identify differences between linear and non-linear mapping methods as well as explore the differences in the processes that affect the DIC between different models. Based on the aforementioned, I recommend the publication of this study. Here are just some suggestions I believe would add value to the publication:

General Comments:

- There is an emphasis throughout the manuscript on the application of this method to reconstruct ocean carbon storage from real-world Argo observations. I was expecting such reconstruction at the end of the paper and a comparison to existing reconstructions (such as GLODAP). I understand that this paper is intended as a "proof of concept" study, followed by another one regarding its "real-world applications". If this is decided as the final form of the study, without including the reconstruction using real Argo measurements, I think the potential application to Argo measurements should be de-emphasized throughout the text.

*The Ensemble Optimal Interpolation approach is a novel approach for ocean carbon in that it is a flexible reconstruction that draws upon linear relationships between the ocean state and carbon content. The synthetic tests that we conduct in this study form a theoretical basis for a future application to real Argo observations. The skill evident in the synthetic reconstructions may be viewed as an upper bound and thus an important aspect for understand the true potential of the method.*

*The expansion of temperature and salinity observations is a key determinant of this reconstruction potential, and we feel that the Argo programme will be a key aspect in the final reconstruction. Additionally, errors from the observations and gridding procedures may be significant, and so we have aimed to understand errors from the covariance fields and errors from the observations separately. As such, we can restrict the Argo discussion and emphasise that this work is an intermediate step, but we do feel that the potential application is an important motivating factor for the work. We have also altered the manuscript to emphasise that the "Argo observations" are synthetic rather than actual observations.*

- The results in this study are focused on the top 100m, however, I couldn't find in the text the motivation to choose such a horizon. Is it because the mixed layer drives the variability in carbon sink? Or, do the statistical relationships between temperature and salinity and DIC break down in the interior ocean?

*Originally, we focused on the top 100m to provide an illustration of how the model covariance fields can be translated into reconstruction skill; however, this decision can be extended with additional decisions (e.g., whether covariance fields are calculated for interior layers or for full integrated carbon). Based on the comments made by both reviewers, we have extended the manuscript by including a section on reconstructing the full depth range covered by Argo-style profilers (0-2000m). We have included example plots and our further analysis in the beginning of our response.*

*We find that the reconstruction skill substantially decreases when moving to depths deeper than 500m, where the ensemble shows more uncertainty and less-independent information in the relationship between DIC, salinity, and temperature (see this document Figure 2, panels c-d and h-i). However, when considering full column DIC, the response is dominated by the responses within the top 500 or 1000m (this document Figure 3, and new manuscript Figure 9).*

- Since it is possible to separate the detrended covariance fields into pCO2 and non-pCO2 terms, if I understood it correctly, it would also be possible to reconstruct the preindustrial DIC and thus calculate the Anthropogenic Carbon fields as the difference between Total and the Preindustrial DIC. If this is possible, it should be mentioned in the paper as possible applications of the method, as there is a significant interest in the community regarding the drivers of the variability of anthropogenic carbon sink (e.g. Gruber et al., 2018: 10.1146/annurev-marine-121916-063407; Gruber et al. 2019: 10.1126/science.aau5153; DeVries 2017: 10.1038/nature21068).

*A preindustrial reconstruction would not necessarily need the breakdown of the covariance fields into terms proportional to pCO2 that we use in Section 3. The full covariance fields could be used to reconstruct preindustrial DIC under two conditions: 1) the covariances between T, S, pCO2, and DIC are stationary even when considering very low emissions of carbon, and 2) there are sufficient T and S observation in the preindustrial to provide information on preindustrial DIC for most of the ocean. Assuming stationary may be somewhat inaccurate, but the lack of observations in the preindustrial period is likely to be a major bottleneck for this application.*

- When reconstructing the carbon time-series from Argo-style observations in Section 6, two observation locations are used, one at year 2002 (with very sparse observations) and another at year 2015 (with many more observations). As far as I understand, the entire time-series (1955-2014) is reconstructed based on the spatial distribution of Argo observation locations in 2002 and 2015, but the number of locations is constant in time. E.g. carbon storage in 1960s is reconstructed from observations that did not exist in the real world. Why not account for both the temporal and spatial distribution of the observation locations? This would give a true insight into the potential of the real-world Argo measurements to reconstruct the time series of carbon storage.

*Yes, when implementing the method with real world data, the reconstruction will be dependent on the number of available observations within a given year. Our original idea behind a constant field of observations as determined by a specific Argo year was to examine how different aspects of carbon variability were represented with different levels of observations, to see if interannual or multidecadal variability required different amounts of nearby observations.*

*To show more easily where a real-world reconstruction might fit inside these limits, we will include a reconstruction using the time-varying distribution of observations that replaces Figure 7 in the original manuscript (Figure 9, new manuscript).*

*The figure expands upon the original Figure 7 in two ways: a) we illustrate reconstructions over the full upper 2000m profiled by Argo floats, and b) we consider the time-varying coverage of Argo profiles up to year 2014 (we refrain from using more recent Argo coverage as the historical experiments run to year 2014). Additionally, to illustrate the potential information provided by the Argo profiles, we provide a comparison of model truth and reconstructed "decadal" DIC changes, calculated as the difference between the 2001-2006 DIC and 2009-2014 DIC. While the intersection between years with Argo coverage and years contained in the historical period is short, the inclusion of these almost-decadal changes provides additional insight as to what sort of information the reconstruction provides on longer timescales.*

Comments by Line:

L150: What does it mean "6 months of profiles"? One profile at least in each of the 6 months? 6 profiles in total?

*We considered a bin observed if, after binning, there was at least one profile taken in 6 unique months of the year (here, we consider the year to run from January to December). We have changed the text to read (Line 148-151):*

> *Any grid cell that has Argo observations for at least 6 months within a given year (running from January to December) is taken to be sufficiently observed, and the modelled annual average profile there is used as an observation for the reconstruction.*

L198: I think there should be a full stop after pCO2 instead of a coma to make the sentence less confusing.

*We have broken the sentence down to explain each set of terms individually.*

L199: Should DIC' and T' be in italics?

*Yes, thank you for pointing this out. The text in line 199 has been adjusted.*

Figure 4: Would it be possible to use the same colorscale? This would make the comparison of the magnitude of the coefficients more intuitive.

*For the top 100m, the coefficients for pCO2, T, and S have the same order of magnitude; however, when moving to the interior ocean the coefficients for S are an order of magnitude larger (Figure 1 in this manuscript). The real impact on the solution for DIC remains small as the variance of salinity is small relative to that for pCO2, for instance. If normalising, the magnitude of coefficients should be considered alongside a comparison of the magnitude for changes in pCO2, T, and S, as it will be the product of the magnitudes that determines their contribution in (3). As these magnitudes are model-specific we have refrained from this normalisation.*

L378: Do you mean Fig 6a and d, instead of a and c?

*Yes, thank you for pointing this out. The reference has been adjusted.*

REFERENCES

Bushinsky, S. M., Landschützer, P., Rödenbeck, C., Gray, A. R., Baker, D., Mazloff, M. R., et al. (2019). Reassessing Southern Ocean air-sea $CO_2$ flux estimates with the addition of biogeochemical float observations. Global Biogeochemical Cycles, 33, 1370– 1388. https://doi.org/10.1029/2019GB006176

Lauderdale, Jonathan M. et al. "Quantifying the Drivers of Ocean-Atmosphere $CO_2$ Fluxes." Global Biogeochemical Cycles 30, 7 (July 2016): 983–99.

Keppler, L., Landschützer, P., Gruber, N., Lauvset, S. K., & Stemmler, I. (2020). Seasonal carbon dynamics in the near-global ocean. Global Biogeochemical Cycles, 34, e2020GB006571. https://doi.org/10.1029/2020GB006571

---

## Author Response (AR2)

**Response to reviewers on "Reconstructing ocean carbon storage with CMIP6 models and synthetic Argo observations"**

To the editor and reviewer:

Thank you for your review to our updated manuscript. The reviewer's comments have been very helpful, and we appreciate how they have considered the methodology as our reconstruction has moved further into the ocean interior. These considerations have made our study more rigorous and a better step towards a real-world carbon analysis.

We have updated the figures to reflect the reviewer's point on de-drifting the data before creating an analysis. There are some improvements in the deeper carbon fields after de-drifting. Overall, the main messages of the previous manuscript remain, as most carbon changes are within the upper part of the water column.

We have attached our responses to the reviewer's comments below.

Best regards,
Katherine Turner

**Authors' response to review**
Reviewer's text in black
Authors' response in blue

I want to thank the authors for their extensive review, their detailed answers to all major and minor comments, and their patience with me when I did not understand everything. The extension of the analysis to the upper 2000 m are a great plus and after the response, I entirely agree that it is best to apply the method to the ARGO data in a separate study.

However, I have two last outstanding major question with respect to the Methods that need to be clarified before publication and that could substantially improve the results. Although I have clicked 'major revisions', I do not want you to see this as a 'traditional' major revisions. The manuscript is in great shape but I think the two major points would make it much better. The first point would have to be adressed in text if the model drift was accounted for and needs additional analysis if it was not. The second point could be adressed by changing the text, but I believe an additional analysis might substantially strengthen the study.

1) I could not find information if the model output was detrended using the pre-industrial control output. Trends are usually small to negligible close to the ocean surface due to exchanges with the atmosphere but can be substantial in the subsurface ocean. If such drifts exist and vary across the models in the ensemble, it might be almost impossible to obtain the optimal weights from these models. Such a drift may well cause the bipolar optimal coefficient pattern for T and S that is shown in Figure 7. If the drift is not accounted for, I think it has to be accounted for. The best way is probably to fit a spline to the pre-industrial control run and to remove it from the historical simulation. In this way, the inter-annual or decadal variability, which are likely similar in the piControl and historical run, is not removed with the drift.

Thank you for this suggestion. We have now included drift removal in the current manuscript as it was indeed lacking from our previous analysis. In the manuscript we have included the line:

> "The drifts in temperature, salinity, and DIC were calculated and removed by subtracting linear trends at each ocean grid cell in the piControl runs." (Section 2.1.1)

Instead of fitting a spline to the piControl runs, we have opted to take the linear trend in ocean temperature, salinity, and DIC at every oceanic grid cell before regridding and integrating vertically into our depth horizons. This drift removal makes minimal assumptions about the nature of drift for any particular model.

We have updated the figures to include this de-drifting, although we do note (from the qualitative similarity between the previous manuscript figures and this manuscript's figures) that model drift does not have a large impact on our results. We speculate that the uncertainties included in the model ensemble outweigh the uncertainties provided by the long-term drifts, and that the process of integrating carbon, temperature, and salinity within relatively thick layers may allow for some compensation between drifts at different depths.

2) It is somehow concerning that problems arise for MPI-ESM1.2-LR because of the higher spatial resolution. I believe that a mistake was made. Following Séférian et al. (2020) (https://link.springer.com/article/10.1007/s40641-020-00160-0), MPI-ESM1.2-LR has one of the coarsest resolutions. If the MPI-ESM1.2-LR resolution is indeed coarser, the argument would have to be altered. At the moment, the weak performance of the DIC reconstruction under MPI-ESM1.2-LR is discussed essential weakness of this method when being applied to observations and regions with lots of mesoscale processes. However, this might be wrong if MPI-ESM1.2-LR was not highly resolved. From my experience, MPI-ESM1.2-LR is more likely an outlier and may not affect the performance of the methods when applied to observations. I believe an adjustment of the Discussion is absolutely necessary. However, I believe the authors could significantly increase trust in the reconstruction method, and I believe the method merits it, by including GFDL-ESM4 (1/2° resolution) and GFDL-CM4 (1/4° resolution). Both model versions are highly resolved compared to other models and GFDL-ESM4 is the best performing model with respect to historical carbon uptake (Annex in Terhaar et al., 2022). It would take a large amount of work and I hence cannot ask for it but I think it would be a worth it.

MPI-ESM1.2-LR is indeed a relatively low-resolution model for the CMIP6 ensemble – thanks for pointing this out. Our statements are incorrect and were originally made for a reconstruction that used the high-resolution MPI ESM. We have modified the text to correct this error.

We have refrained from using the GFDL models in this proof-of-concept work as the models have only one realization. For this step we restricted our study to only models that had multiple realizations (5) in order to include uncertainties from both model architecture and initial conditions/ the phasing of climate modes of variability. As we continue our work and move towards creating a real-world reconstruction, we will be updating our ensembles to include other models such as those from GFDL and the NCC (as NorESM is provided as a test case but not included in the original model ensemble).

In addition, I have one comment that is neither major nor minor:

1) Section 3.1 seems to be complicating the message. The co-evolving trends of atmospheric CO2 and warming lead to correlations that are not related to the effect of warming on the ocean system, i.e., changes in circulation and solubility. In fact, the ensemble coefficients in Figure 4 show this as the positive correlation from temperature is entirely accounted for by the increase in pCO2 and the pCO2 coefficient, whereas the T coefficient accounts for the solubility. I believe that Figure 4 really shows the strength of the approach and merits to be a centerpiece of the results.
I am not an author of this study but would seriously consider removing section 3.1, as it confused me more than it helped. However, I am only one person and others may disagree. And as nothing is wrong with the section, I only wanted to voice my opinion and do not want to ask for changes.

We have opted to include the sections on the correlations and the breakdown of the correlations between DIC and temperature to tie in more clearly to our future work. These synthetic reconstructions assume that observations taken from the model output are perfect. Therefore, the least squares solutions shown in Figure 4 is the optimal solution. The optimal coefficients will change when including imperfect observations, but the covariance fields used by the Ensemble Optimal Interpolation algorithm will remain the same if the ensemble makeup is the same. Thus, while the inclusion of the correlation fields makes the paper longer, we feel it is an important step to show for future work. We have included a statement in the beginning of 3.1 that reflects our logic in including the covariance fields first.

Please find below some minor comments:

1) Figure 8: It seems that the reconstruction performs best, where most carbon is stored (North Atlantic and Southern Ocean) and performs poorly where little to no carbon is stored, for example in the deep North Pacific. Thus, the maps in Figure 8 might suggest that the reconstruction performs poorer than it does. Maybe it would be better to show differences between the reconstructed and the true DIC, like Figure 9.

The statistics in Figure 8 have been provided to show areas where uncertainties in the models could lead to conservative estimates of DIC as well as regions that remain difficult to predict due to errors in the covariance fields. Because of the multiple models used, it would be difficult to isolate specific difference plots. However, we have included an additional figure in the Supplementary that shows the standard deviation of DIC' across the ensemble, as well as the ensemble average RMSE from the individual reconstructions. This figure clearly reflects the ideas you have mentioned, where the model does well in regions with high DIC' changes (and thus a high standard deviation), while the regions with error increases generally have low DIC' changes.

2) Line 3 of the abstract: I would maybe highlight here that the temperature and salinity coverage is much less sparse than the carbon observation coverage. The contrast between both coverages is what this is all about.

Thank you, we have extended the sentence to emphasize this point.

3) Line 20: Should probably be Terhaar et al. (2022) and not 2020.

We have revised the manuscript to reflect the correct citation.

4) Line 28: would replace 'sufficient' by 'enough'. The coverage of, 1.5% is likely not sufficient (L. Gloege, G. A. McKinley, P. Landschützer, A. R. Fay, T. L. Frölicher, J. C. Fyfe, T. Ilyina, S. Jones, N. S. Lovenduski, K. B. Rodgers, S. Schlunegger, Y. Takano, Global Biogeochem. Cycles, in press, doi:https://doi.org/10.1029/2020GB006788.)

We have updated this sentence to describe the order of magnitude of $pCO_2$ observations rather than qualifying with "sufficient."

5) Lines 38 to 47: This paragraph should probably also include reference to the application of the eMLR* method by Gruber et al. (2019) (https://www.science.org/doi/10.1126/science.aau5153)

We have included this work with an emphasis that these reconstructions are concerned with the evolving anthropogenic carbon inventory.

6) Lines 56-57: What exactly are these well-understood relationships? Maybe worth to elaborate a bit more.

We have split up this sentence to more explicitly describe the inverse relationship between temperature and DIC and the direct relationship between salinity and DIC.

7) Line 59: Maybe worth to reference Weiss et al. (1974). (https://www.sciencedirect.com/science/article/pii/0304420374900152)

We have included this citation in the first part of the expanded relationships section above.

8) Line 234-235: I believe that this sentence is not correct. Fig. 2 is rather a combination of the parallel increase in atm CO2 and temperature and the solubily effect of T on pCO2.

We have altered the sentence to read: "Thus the heterogenity found in the overall correlation between DIC and temperature in Figure 4 can be understood as the sum of an emissions-driven undersaturated response that correlates to but is not driven by warming, and a temperature-driven solubility response."

9) Line 336: How many regions are 'most regions'.

We have extended the sentence to read "Across the sensitivity tests and outside of the Southern Ocean and small regions in the North Atlantic, North Pacific, and equatorial Pacific, carbon can be reconstructed with a relative RMSE reduction of at least 50%."